# Neural mechanisms for selectively tuning in to the target speaker in a naturalistic noisy situation

Bohan Dai [1,2,3], Chuansheng Chen[4], Yuhang Long[1], Lifen Zheng[1], Hui Zhao[1], Xialu Bai[1], Wenda Liu[1], Yuxuan Zhang[1], Li Liu[1], Taomei Guo[1], Guosheng Ding[1] & Chunming Lu[1]

The neural mechanism for selectively tuning in to a target speaker while tuning out the others in a multi-speaker situation (i.e., the cocktail-party effect) remains elusive. Here we addressed this issue by measuring brain activity simultaneously from a listener and from multiple speakers while they were involved in naturalistic conversations. Results consistently show selectively enhanced interpersonal neural synchronization (INS) between the listener and the attended speaker at left temporal–parietal junction, compared with that between the listener and the unattended speaker across different multi-speaker situations. Moreover, INS increases significantly prior to the occurrence of verbal responses, and even when the listener's brain activity precedes that of the speaker. The INS increase is independent of brain-to-speech synchronization in both the anatomical location and frequency range. These findings suggest that INS underlies the selective process in a multi-speaker situation through neural predictions at the content level but not the sensory level of speech.

[1] State Key Laboratory of Cognitive Neuroscience and Learning & IDG/McGovern Institute for Brain Research, Beijing Normal University, Beijing 100875, China. [2] Max Planck Institute for Psycholinguistics, Nijmegen 6525 XD, The Netherlands. [3] Donders Institute for Brain, Cognition and Behavior, Radboud University, Nijmegen 6525 EN, The Netherlands. [4] Department of Psychology and Social Behavior, University of California, Irvine 92697-7085 CA, USA. Correspondence and requests for materials should be addressed to C.L. (email: luchunming@bnu.edu.cn)

I n a noisy, crowded environment, a listener can still tune into the speech of the target speaker while tuning out the others[1]. This remarkable ability – the so-called "cocktail-party" effect[2] – has been studied for over half a century. However, the neural mechanism underlying this selective process is still not well-understood.

Two lines of previous research have implicated the synchronization of neural activities as a potential neural mechanism for the selective processing of information in a multi-speaker situation. Previous studies have found associations between the movement of articulators and brain activity in the speaker (brain-to-articulation synchronization)[3], and between the temporal features of the speech sound and brain activity in the listener (brain-to-speech synchronization)[4–12]. Although it has been speculated that there should be a relationship between the brain activity of the listener and that of the attended speaker (i.e., brain-to-brain synchronization) in a multi-speaker situation, this hypothesis has never been tested. Brain-to-brain synchronization can be indexed by interpersonal neural synchronization (INS, i.e., brain activities from two persons covary along the time course). INS has been found to be associated with various aspects of communication behaviors such as facial expressions and turn-takings between interlocutors[13]. INS was also related to better prediction and comprehension when the recorded speech was played to the listener (when the speaker and listener were scanned sequentially using functional magnetic resonance imaging (fMRI)[14]. Moreover, recent evidence additionally demonstrated that such brain-to-brain synchronization is independent of brain-to-speech synchronization in a clear auditory situation[15]. Thus far, however, no studies have tested the potential role of INS in the cocktail-party effect.

Thus, it remains unknown whether INS underlies the selective processing of the target information in a multi-speaker situation. Is INS between the listener and the attended speaker (the LA pairs) selectively enhanced to a greater degree than that between the listener and the unattended speaker (the LU pairs)? Which level of information (i.e., sensory-level or high-level) is related to the enhanced INS? Given the previous findings on the role of neural synchronization in speech comprehension both within and between brains, we expected a greater INS increase for the selective process. Moreover, previous evidence indicates that the brain will generate internal predictions to optimize behavior[16]. In a conversation, the listener uses content predictions to determine what to say (high-level information) but not when to say it (sensory-level information)[17]. It was consequently hypothesized that the neural activity associated with content prediction would be higher prior to a response than at other time points, but the neural activity associated with sensory-level information would not show that pattern. This hypothesis has been supported by recent neural evidence in a non-conversational context[18]. Based on this piece of evidence, we would be able to examine which level of information was associated with selectively enhanced INS in a multi-speaker situation.

To address the above research questions, the current study employed functional near infrared spectroscopy (fNIRS)-based hyperscanning to examine INS in a naturalistic multi-speaker situation. Although fMRI and electroencephalography can measure brain activities simultaneously from multiple participants (i.e., hyperscanning)[19], they cannot be used to study naturalistic multi-speaker conversations[20]. In contrast, fNIRS has clear advantages such as portability, high tolerance of movement artifacts, and measurements of local hemodynamic effect[21–23]. Using fNIRS-based hyperscanning, we expected to identify INS correlates of the selective process and to determine the level of information with which INS was associated. The results show selective enhancement of INS between the listener and the attended speaker compared with that between the listener and the unattended speaker across different multi-speaker situations. Moreover, the selectively-enhanced INS is closely associated with the processing of the high-level content information rather than the sensory-level information.

## Results

**Participants and procedures**. Sixty-six adults were pseudorandomly split into 22, 3-person, same-sex groups, but one group did not yield valid data due to technical failures, resulting in a final sample of 63 adults from 21 groups. For each group, one of the three participants was randomly assigned as the listener,

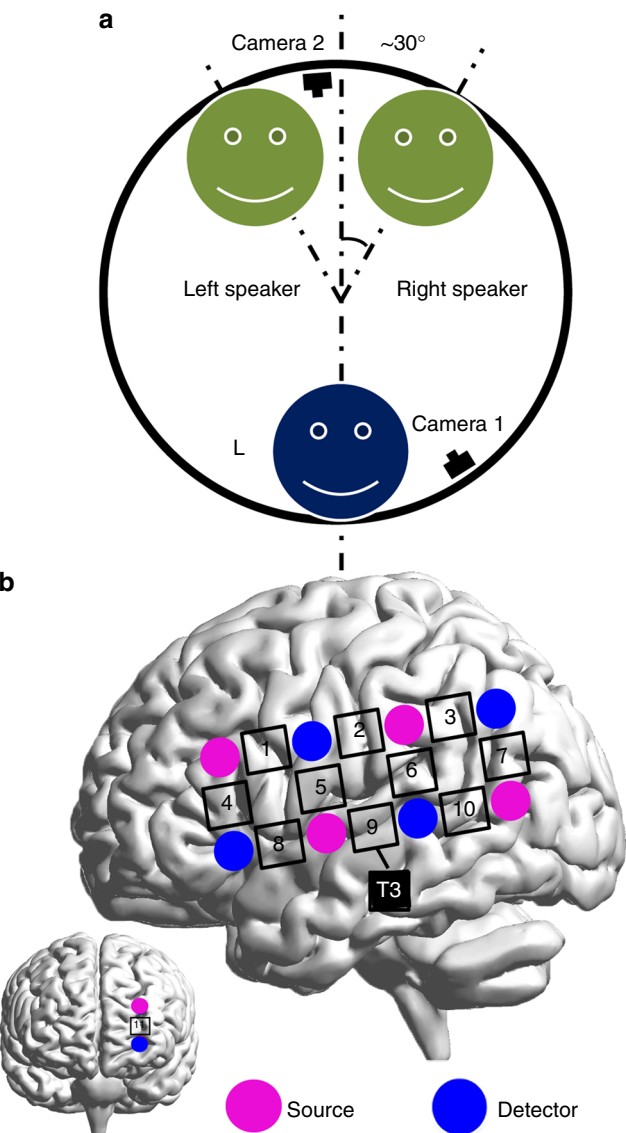

**Fig. 1** The setup of the experiment. **a** For each group, three participants sat in a triangle. As viewed by the listener (L), the locations of the left and right speakers were −30° (left) and +30° (right), respectively. One camera was placed to the right (from the listener's perspective) of the left speaker, whereas the other camera was placed to the right of the listener. Participants performed the tasks in a face-to-face condition and a back-to-back condition. L indicates the listener. **b** The optode probe set was placed on left frontal, temporal, and parietal cortices. CH9 was placed just at T3 in accordance with the international 10–20 system. The brain images were automatically generated using the software of BrainNet Viewer, which is freely available on the NITRC[51]

whereas the other two participants as the speakers (Fig. 1a and Supplementary Figure 1). During the experiment, a baseline session (i.e., resting-state with eyes closed) was followed by two task sessions: a face-to-face condition and a back-to-back condition. Each group performed three communication tasks within each of the two experimental conditions. One communication task was excluded (see Methods section and Supplementary Methods for the exclusion of one task). The two tasks used in this study were as follows: (1) only one speaker talked to the listener while the other speaker kept silent; and (2) two speakers talked simultaneously while the listener was required to attend only to one speaker throughout the task (see Methods section). The order of the face-to-face and back-to-back conditions was counterbalanced across the participant groups. All three participants were video-recorded during the experiment and their communications were subsequently coded for timing and content (see Methods section). Moreover, immediately after each task, the participants were asked to assess the quality of their communication on a five-point scale (see Methods section).

FNIRS was used to collect brain functional data from the left hemisphere of all three participants simultaneously (Fig. 1b). Wavelet transform coherence (WTC) was used to assess the cross-correlation between the two fNIRS time series of $11 \times 11$ combinations of measurement channels (CH) generated by pairs of participants as a function of frequency and time. The frequency-averaged coherence value between 0.1 and 0.4 Hz was calculated and then further averaged across the time points (see Methods section). The averaged coherence value in the resting-state session was subtracted from that of the task session, and the difference was used as an index of the INS increase for each pair of the participants. Three $11 \times 11$ matrices of the INS increase (for the three dyadic pairs within each group) were generated for each task. To have an overall picture of the INS increase, three-way repeated measures analysis of variances (ANOVAs), i.e., 2 (face-to-face vs. back-to-back) × 2 (single-speaker task vs. multi-speaker task) × 2 (LA vs. LU), were conducted with False Discovery Rate (FDR) correction for all CH combinations ($P <$ 0.05). To further examine the significant main and interaction effects, pairwise comparisons were conducted by using two-sample $t$-tests with FDR correction for all CH combinations ($P <$ 0.05, two-tailed). To determine either the increase or decrease of INS for a specific CH combination, one-sample $t$-tests with FDR correction ($P < 0.05$, two-tailed) on the INS increase of the LA, LU, and AU (attended and unattended speaker) pairs were also conducted. Finally, correlations between the INS increase and communication behaviors and communication quality were

obtained in order to test which level of information was associated with the INS increase.

**Selective enhancement of INS only in a multi-speaker situation.** ANOVAs showed a significant main effect for the LA vs. LU comparison, with a significant difference only in their INS increase for the temporal-parietal junction (TPJ, CH3) of the speaker and that of the listener (ANOVA, $F(1, 20) = 42.097$, $P <$ 0.001). No other significant effects (either main effects or interactions) were found (ANOVA, $P > 0.05$). To explicate the omnibus ANOVA results and to further examine the patterns of INS for each type of tasks (multi-speaker and single-speaker) and each condition (face-to-face and back-to-back), we conducted the following one-sample or two-sample $t$-tests.

We first examined the data from the multi-speaker tasks. To ascertain the comparability of the verbal and non-verbal inputs between the LA and LU pairs, patterns of verbal and non-verbal communication behaviors from the two speakers were subjectively assessed by the listener and objectively analyzed by two additional coders (based on the experimental video) (see Supplementary Methods). No significant differences were found between the two speakers (two-sample $t$-test, $P > 0.05$, two-tailed). Next, the increase of INS was examined for the multi-speaker tasks for each condition. As expected, INS showed selective enhancement (Fig. 2), with the brain activity at TPJ-TPJ (CH3-CH3) having significant INS increase for the LA pairs under both the face-to-face and back-to-back conditions. No significant INS increase was found for the LU and the AU pairs for either the face-to-face or the back-to-back condition. In addition, the face-to-face condition had an additional increase of INS between pSTC (CH7) of the listener and TPJ of the attended speaker (pSTC-TPJ, CH7-CH3) (Fig. 2). No such effects were found in the back-to-back condition.

The differences between the LA and LU pairs were directly tested by a two-sample $t$-test across all CH combinations with FDR correction at $P < 0.05$ level (two-tailed). Results showed that the INS increase at TPJ-TPJ (CH3-CH3) was significantly stronger for the LA pairs than for the LU pairs (two-sample $t$-test, face-to-face: $t(20) = 4.049$, $P < 0.001$, two-tailed; back-to-back: $t(20) = 3.756$, $P < 0.001$, two-tailed) (Fig. 3a, b). Compared with the back-to-back condition, the face-to-face condition had an additional difference between the LA and LU pairs in the increase of INS at pSTC-TPJ (CH7-CH3) (two-sample $t$-test, $t(20) = 3.878$, $P < 0.001$, two-tailed). No other significant differences were found between the LA and LU pairs for other CH combinations.

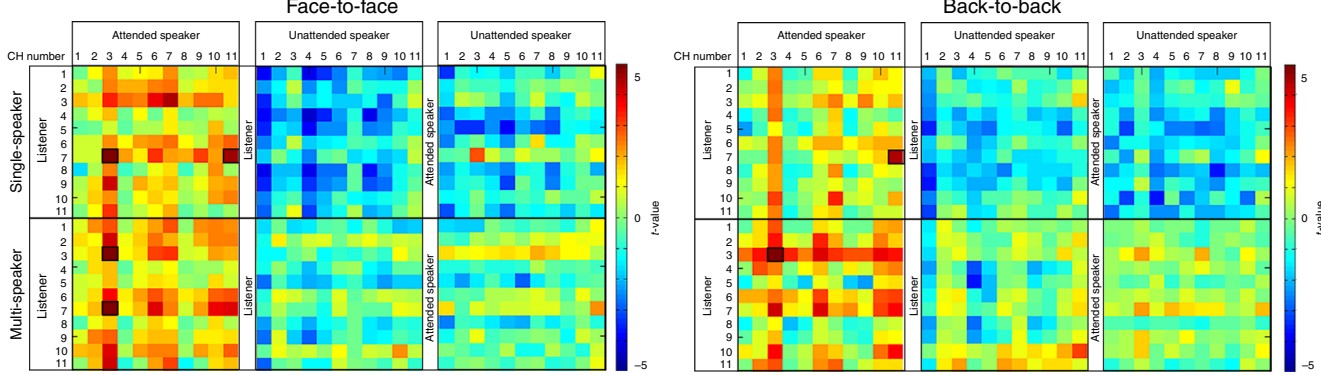

**Fig. 2** The increase of INS in each pair of participants for each task and condition. INS was calculated for all CH combinations, generating three $11 \times 11$ matrices (for the LA, LU, and AU pairs) of $t$-values for each task within each condition (one-sample $t$-test, two-tailed). The 2 tasks were the single-speaker task and multi-speaker task. The significant CH combinations are highlighted by black squares

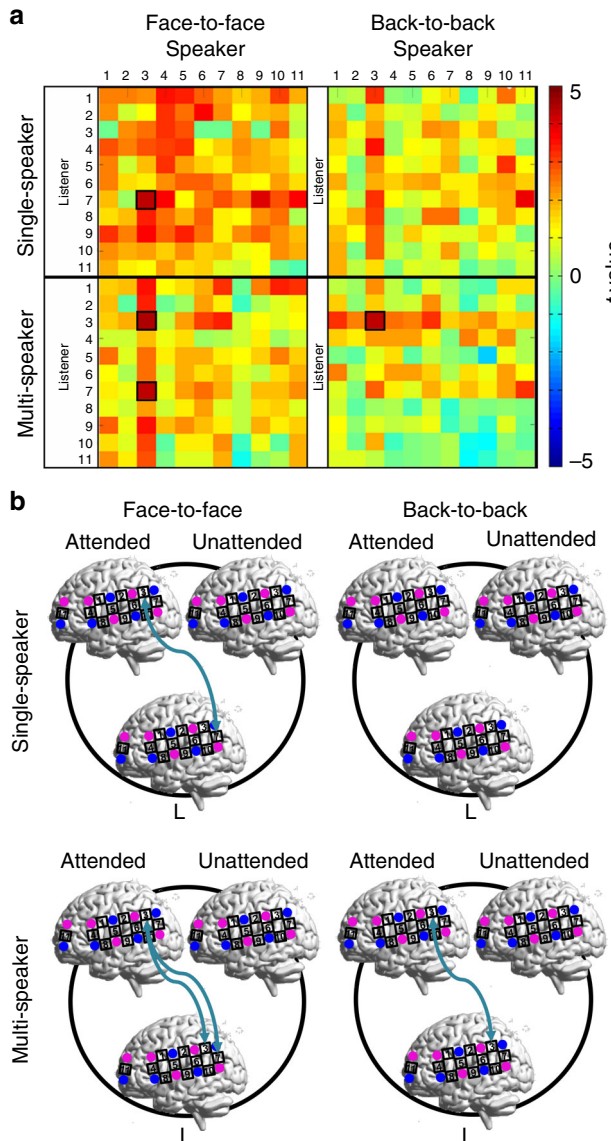

**Fig. 3** Two-sample *t*-test between the LA and LU pairs. **a** Differences between the LA and LU pairs in the increase of INS are shown for all tasks by condition. The increase of INS was thresholded at $P < 0.05$ after FDR correction for all CH combinations. **b** The thresholded results are displayed as brain-to-brain connections (arrows in turquoise). L indicates the listener, and Attended and Unattended indicate the respective speakers. The brain images were automatically generated using BrainNet Viewer, which is freely available on the NITRC[51]

As would be expected for a naturalistic multi-speaker situation, the listener in this study sometimes responded to the speaker and spoke continuously (about 10% and 11% of the 280 s on average in the face-to-face and back-to-back conditions, respectively). To exclude the potential confounding effect of the listener's continuous speech on the increase of INS, the INS increase was re-calculated based on time points when the listener was silent. The new analysis confirmed significant differences between the LA and LU pairs at TPJ-TPJ (two-sample *t*-test, CH3-CH3, face-to-face: $t(20) = 4.126$, $P < 0.001$, two-tailed; back-to-back: $t(20) = 3.855$, $P < 0.001$, two-tailed) and pSTC-TPJ (two-sample *t*-test, CH7-CH3, face-to-face only: $t(20) = 4.056$, $P < 0.001$, two-tailed).

Second, we analyzed the data from the single-speaker tasks. Results showed no significant INS increase at TPJ-TPJ (CH3-

**Table 1 Prediction accuracy of the speakers' identity based on the increase of INS**

| Conditions | Tasks | LA (%) | LU (%) | Total (%) |
|---|---|---|---|---|
| Face-to-face | Single-speaker task | 95 | 95 | 95 |
| | Multi-speaker task | 76 | 91 | 83 |
| Back-to-back | Single-speaker task | 91 | 91 | 91 |
| | Multi-speaker task | 71 | 86 | 79 |

CH3). Instead, two significant INS increases were found between pSTC (CH7) of the listener and two CHs (TPJ, CH3; prefrontal cortex (PFC), CH11) of the attended speaker (Fig. 2). No significant INS increase was found for the LU pairs (the unattended "speaker" was actually silent because it was a single-speaker task) or the AU pairs. Direct comparisons between the LA and LU pairs showed a significant difference at pSTC-TPJ (CH7-CH3, two-sample *t*-test, $t(20) = 3.314$, $P < 0.001$, two-tailed), but the difference at pSTC-PFC (CH7-CH11) did not survive the FDR correction across all CH combinations (two-sample *t*-test, $P > 0.05$, two-tailed). No other significant differences were found (Fig. 3).

**Validation of the selective enhancement of INS through permutation.** To verify that the INS increase was specific to the original pairing of the listener and the attended speaker, a validation approach (i.e., a permutation test for 1000 times) was used. For each multi-speaker task, all 63 participants were randomly assigned to 21 new 3-person groups, and then the INS analysis was re-conducted. The results did not show any significant INS increase for any types of pairings (Supplementary Figure 2). These results suggest that the significant INS increase was specific to the original LA pairs.

**Prediction of the attended speaker based on the increase of INS.** Although the univariate statistics described above revealed an important role of TPJ (CH3) in the selective process in a multi-speaker situation, an alternative analytic strategy is to use the INS data to classify the LA vs. LU pairs. This multivariate approach can be used to confirm the above univariate results, to investigate whether data from other brain areas provided additional information for classification, and to ascertain how accurately the INS data could predict the LA vs. LU pairs. Fisher Linear Discriminate Analysis (FLDA) was used. Specifically, the INS increases for all CH combinations averaged across the whole time courses were used as classification features, and the LA and LU pairs were used as class labels. A leave-one-out cross-validation approach was employed. Results showed that the averaged accuracy reached 83% for the LA pairs and 90% for the LU pairs (total accuracy = 87%) across all tasks (Table 1).

For each task, a different set of CH combinations showed significant contributions to the prediction. However, only the CH combination at TPJ-TPJ (CH3-CH3) was shared for the multi-speaker tasks but not for the single-speaker tasks, suggesting that TPJ played an essential role in a multi-speaker situation.

Finally, when the classification features in the multi-speaker tasks were applied to the prediction in the single-speaker tasks, the accuracy was only 68% on average. These findings additionally suggest that the INS increase was specific to the multi-speaker situations, and this effect was specifically associated with TPJ (CH3).

**Level of information associated with the increase of INS.** To test whether the selective enhancement of INS was associated

with the sensory-level or high-level information, three procedures were conducted.

First, previous evidence shows that during a conversation, the listener uses content prediction to determine what to say but not when to say it[17]. Thus, the neural activity that is associated with content prediction would be higher prior to a response than that at any other time points. To test this hypothesis in INS, we examined the associations between the fluctuations of the INS increase and the time course of communication behaviors. Verbal (e.g., turn-takings and interjections) and non-verbal responses (e.g., orofacial movements, facial expressions, and body gestures) during communications were coded by two coders (see Supplementary Methods and Supplementary Table 1 for behavioral data). Because the INS increase associated with a multi-speaker situation was localized to TPJ (CH3), we examined the time course of the INS increase at this site specifically. The INS increase that ranged from 10 s before the response to 10 s after the response were selected as epochs, and averaged across the same category of epochs to obtain two indexes: i.e., the INS increase of verbal responses and that of non-verbal responses. Meanwhile, the INS increase corresponding to time points where no responses were made (i.e., one person was talking) were also averaged to obtain an index of no responses (for details, see Methods section and Supplementary Figure 3). The INS increases of verbal and non-verbal responses were compared with those of no responses in a point-by-point manner using paired two-sample $t$-tests ($P < 0.05$, two-tailed, FDR correction). Results showed that greater INS increase preceded verbal responses, reaching a peak value around 5–6 s earlier than the occurrence of verbal responses, for both the face-to-face and back-to-back conditions (Fig. 4a). No significant INS increase was found to parallel or follow verbal responses. Also, no significant INS increase was found around non-verbal responses. Because an INS increase was found at pSTC-PFC in the single-speaker tasks only, pSTC-PFC was selected as a control CH combination in order to test whether the effect at TPJ-TPJ was specific to the noisy situation. Moreover, this control was also used to tease apart different high-level cognitive functions such as TPJ-related mentalizing[24] and PFC-related attention, planning, and emotion[25,26]. However, no significant results were obtained for pSTC-PFC (two-sample $t$-test, $P > 0.05$, two-tailed, FDR correction) (Fig. 4d).

In addition, we correlated the INS increases at TPJ-TPJ for each time point along the time course (before verbal responses) with scores of communication quality between the listener and the attended speaker (see Methods section). Results showed significant correlations (Fig. 4b, c). Moreover, the correlation reached a peak value at about the same time as did the INS increase (around 4 s before the occurrence of verbal responses, Pearson correlation, $r = 0.634$, $P = 0.002$). The increase of INS that paralleled or followed verbal responses and those around non-verbal responses did not correlate significantly with scores of communication quality (Pearson correlation, $P > 0.05$, Šídák correction) (Fig. 4b, c). Also, the increase of INS did not correlate significantly with either speaking patterns (speaking rate, percent of disfluency) or frequency of responses in the listener (Pearson correlation, $P > 0.05$, Šídák correction). The same analyses were conducted with the INS increase at pSTC-PFC, and no significant correlations were found (Pearson correlation, $P > 0.05$, Šídák correction) (Fig. 4e, f). These results suggest that the selective INS increase was associated with high-level content information processing that involved TPJ.

Second, several studies[14,27] have found that the strongest INS between a listener and a speaker occurred at a lag in the time courses, probably due to the listener's prediction of the upcoming information. To test this effect in a multi-speaker situation, the time course of the listener was shifted forward relative to that of the speaker and vice versa from 1 to 10 s (step = 1 s). Results showed that the highest increase of INS occurred when the listener's brain activity preceded that of the speaker by 1–3 s (LA vs. LU, two-sample $t$-test, $P < 0.05$, two-tailed, FDR correction, Fig. 5). No other significant results were found at other time-lags or when the speaker's brain activity preceded that of the listener. This result further excluded the possibility that the current INS increase was due to tracking the acoustic speech signal or the visual articulatory movements.

Third, to test the possibility that brain-to-speech synchronization contributed to brain-to-brain synchronization, brain-to-speech synchronization and its relationship with brain-to-brain synchronization were examined in the multi-speaker situations. Recent BOLD-effect evidence indicates that higher frequency fluctuations in BOLD signals (up to 0.8 Hz) have neural relevance and make functional contribution[28,29]. It is known that in BOLD signals, while low-frequency fluctuations are associated with general excitability[30] and spatial overlap of the neural networks[31], high-frequency fluctuations are associated with focal functions and can be a more direct and precise index of the cognitive processes[28]. Moreover, evidence shows reconstruction of real-life sounds from fMRI response patterns (<~2 Hz) in the human auditory cortex[32]. Thus, it is possible to evaluate brain-to-speech synchronization using the fNIRS signal, which also measures the hemodynamic effect as the BOLD signal does.

As there were no significant results after FDR correction across all measurement CH-speech pairs over the full frequency ranges (11 pairs in total, 0.01–0.8 Hz) at $P < 0.05$ level, uncorrected results at a significance level of $P < 0.0005$ were reported (two-sample $t$-test, two-tailed). As expected, only the LA pairs had significant brain-to-speech synchronization at CHs that covered the left sensorimotor and auditory cortices in both the face-to-face and back-to-back conditions. Specifically, in the face-to-face condition, the highest level of brain-to-speech synchronization in the LA pairs appeared at CH5 and CH8–10 that covered the left sensorimotor and temporal auditory cortices at a frequency range of around 0.8 Hz (two-sample $t$-test, CH5: $t(14) = 4.425$, $P = 0.00049$, two-tailed; CH8: $t(14) = 4.45$, $P = 0.00047$, two-tailed; CH9: $t(14) = 4.862$, $P = 0.00021$, two-tailed; CH10: $t(14) = 4.917$, $P = 0.00019$, two-tailed). No CHs survived the thresholding at any frequency ranges in the LU pairs (two-sample $t$-test, $P > 0.0005$, two-tailed) (Fig. 6a). In the back-to-back condition, again, CH5 at the sensorimotor and temporal auditory cortices had significant brain-to-speech synchronization at the same frequency range as that in the face-to-face condition (i.e., around 0.8 Hz) (two-sample $t$-test, $t(14) = 4.618$, $P = 0.0004$, two-tailed). The back-to-back condition had an additional result at CH1 that covered the left inferior frontal cortex (IFC) at a lower frequency range of around 0.05 Hz (two-sample $t$-test, $t(14) = 4.569$, $P = 0.00044$, two-tailed) (Fig. 6a). No significant results were found in the LU pairs. Also, no significant difference was found between the LA and LU pairs (two-sample $t$-test, $P > 0.0005$, two-tailed). These results confirmed previous reports based on EEG/MEG techniques that the activity in the auditory cortex selectively tracks the attended speech while ignoring the unattended one. However, such an effect did not overlap with brain-to-brain synchronization in both the anatomical location and frequency range of the brain activity (Fig. 7).

In addition, significant brain-to-speech synchronization was also found between the brain activity of the speaker and her/his speech. The results appeared at CH4 that covered IFC at the same frequency range as that in the listener's brain (i.e., around 0.8 Hz) in both the face-to-face condition (two-sample $t$-test, $t(14) = 4.41$, $P = 0.00049$, two-tailed) and the back-to-back condition (two-sample $t$-test, $t(14) = 4.606$, $P = 0.00039$, two-tailed)

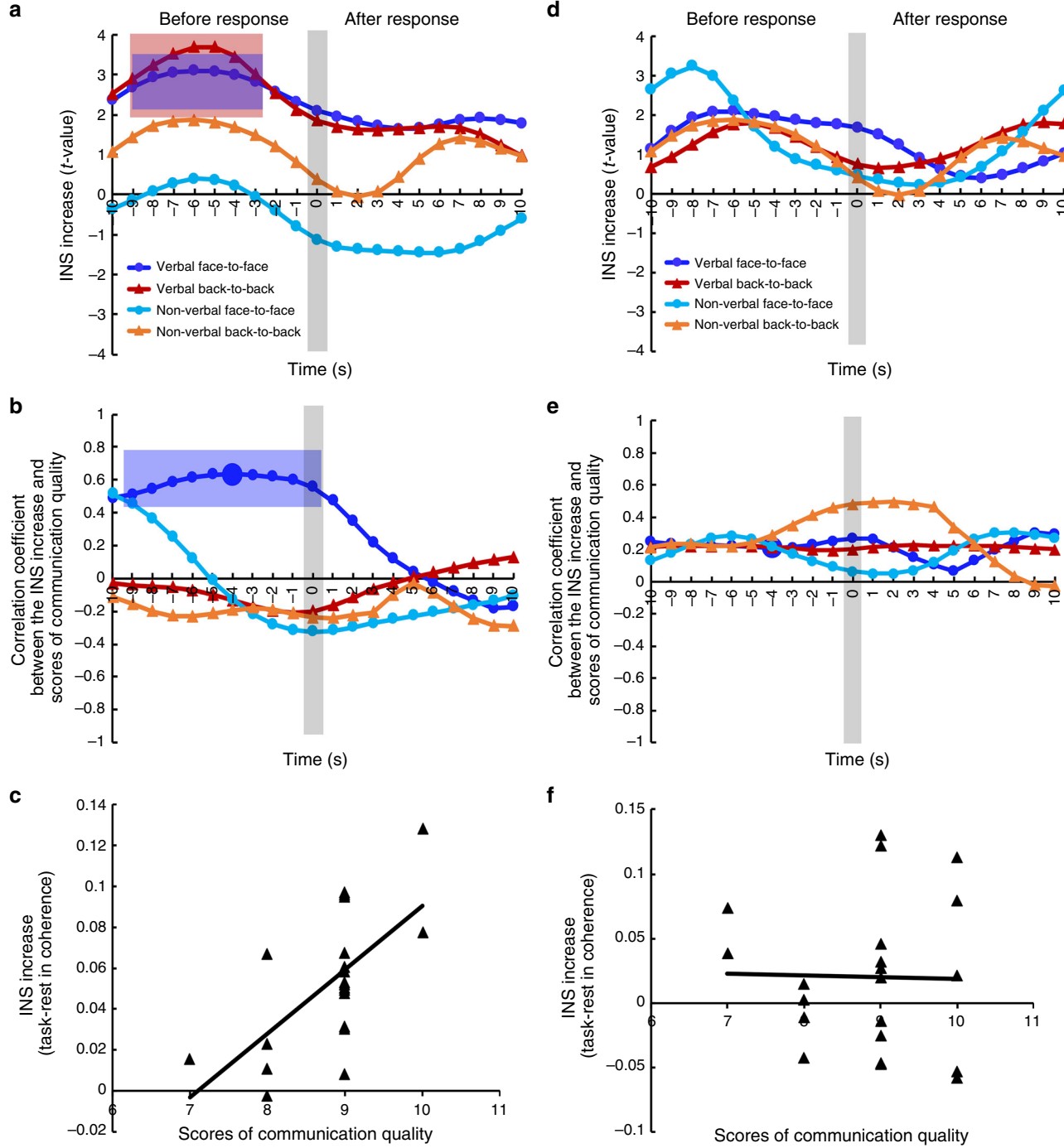

**Fig. 4** Linking the time course of the INS increase with the behavioral data. **a** Time course of the INS increase from −10 s (before response) to +10 s (after response). The 0 s corresponds to the occurrence of responses, which is highlighted by a gray bar. The *y*-axis shows the *t*-value (i.e., two-sample *t*-test on the INS increase, two-tailed). Statistically significant time points are highlighted by colored rectangles. The colors of the rectangles correspond to the colors of the lines. **b** Correlation coefficients between the INS increase and scores of communication quality. Significant correlations are highlighted by a blue rectangle. The peak value of the coefficient is shown as a big circle, and the scatter plot is shown in **c**. **d**–**f** Results on the control CH combination (i.e., pSTC-PFC). The same time point as in **c** is also highlighted by a large circle in **e**, and the scatter plot is shown in **f**. Note that no significant results were obtained for pSTC-PFC

(Fig. 6b). Again, TPJ (CH3) did not show any significant brain-to-speech synchronization in the speakers.

To further exclude the potential influence of these CHs' brain-to-speech synchronization on brain-to-brain synchronization at TPJ, the INS increases for CH combinations between CH4 and all other eleven CHs were averaged. Similar procedures were conducted on CH5 and CH8–10. These averaged INS increases were used as the covariates when comparing the INS increase of TPJ-TPJ (CH3-CH3) between the LA and LU pairs using a repeated measures ANCOVA. Results still showed significantly higher INS increase in the LA pairs than in the LU pairs in the face-to-face condition (ANCOVA, $F (1, 15) = 7.056$, $P = 0.018$).

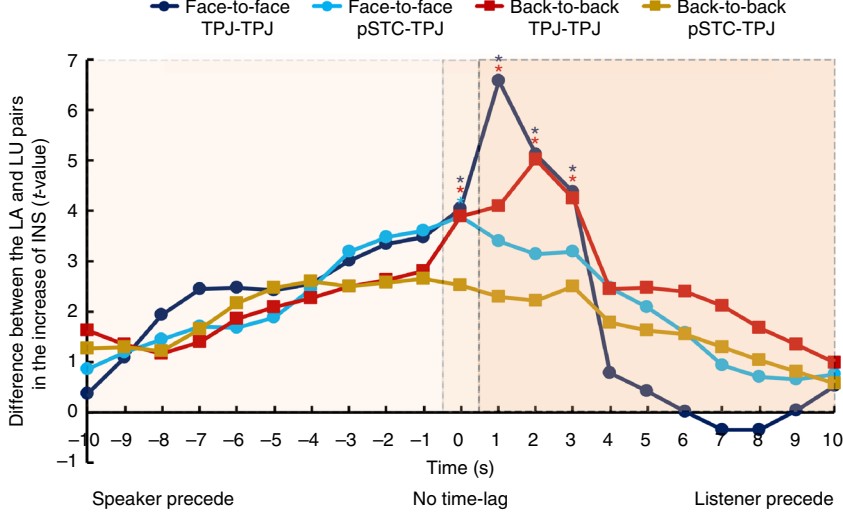

**Fig. 5** Results of the time-lag analysis. The x-axis represents time-lags when the speaker's brain activity preceded that of the listener (from −10 s to −1 s, Speaker precede) and vice versa (from 10 s to 1 s, Listener precede). The 0 s represents no time-lag. The y-axis represents results of two-sample t-test (i.e., LA vs. LU) on the increase of INS (two-tailed). Dark blue and red curves indicate the results at TPJ-TPJ in the face-to-face and back-to-back conditions, respectively. Turquoise and brown lines indicate the results at pSTC-TPJ in the face-to-face and back-to-back conditions, respectively. The stars indicate significant differences between the LA and LU pairs. Note that the result at the time-lag of 0 s is the same as in Fig. 3

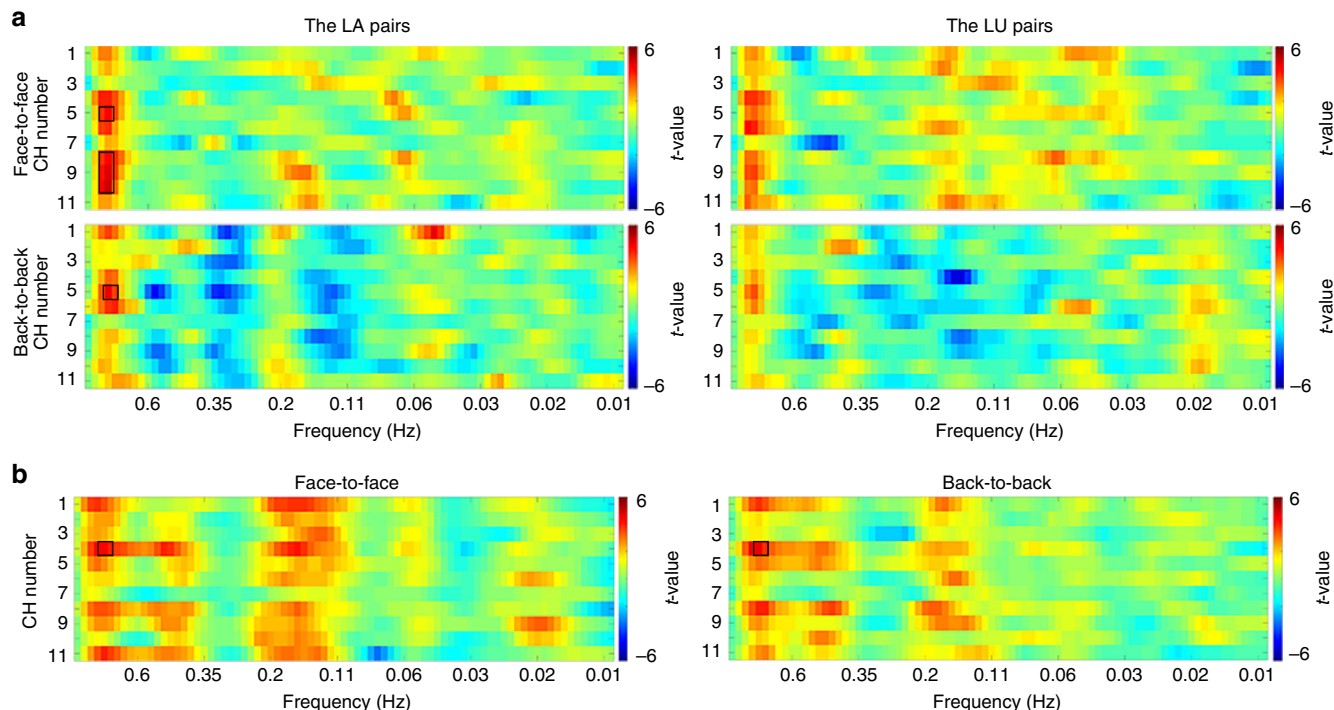

**Fig. 6** T-values of brain-to-speech synchronization. **a** Synchronization between the brain activity of the listener and the amplitude envelope in the attended speech (left) and unattended speech (right) in both the face-to-face (top) and back-to-back (bottom) conditions. **b** Similar analysis between the brain activity of the speaker and the amplitude envelope of her/his speech. Please note that because the left and right speaker were randomly assigned, their data were combined. Two-sample t-test, P < 0.0005, two-tailed, uncorrected. Values that survived the thresholding are highlighted by black squares or rectangles. For each subplot, the x-axis represents frequency in Hz, and the y-axis represents the CH number

Also, in the back-to-back condition, even after adding the covariates, i.e., the INS increase related to CH1, CH4, CH5, CH8, and CH10, the INS increase at TPJ-TPJ (CH3-CH3) was still significantly higher in the LA pairs than in the LU pairs (ANCOVA, $F(1, 15) = 5.588$, $P = 0.032$). In sum, these results excluded the possibility that the selectively enhanced INS at TPJ was contributed by brain-to-speech synchronization either in the listener or in the speaker.

## Discussion

Using a fNIRS-based hyperscanning approach, this study examined whether and how INS underlies the cocktail-party effect. A naturalistic multi-speaker situation was created by having a listener interact with two speakers who produced competing speech. Results showed that the increase of INS (relative to the baseline) was significantly higher in the LA pairs than in the LU pairs. Moreover, the increase of INS peaked before the occurrence of

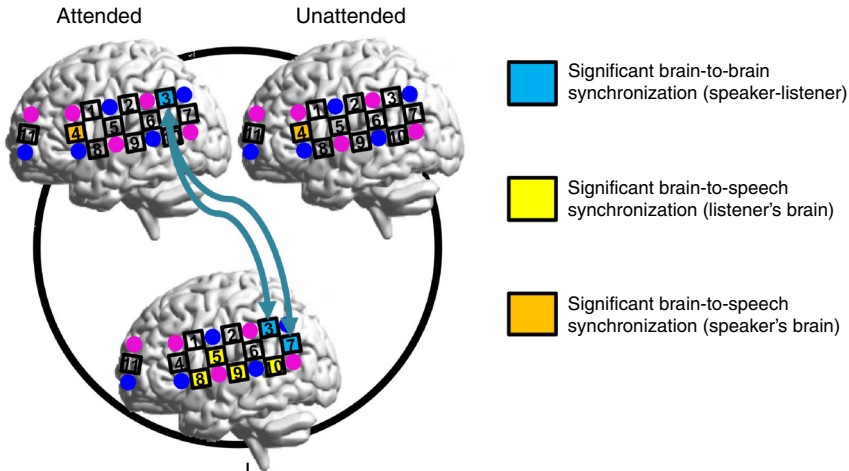

**Fig. 7** Anatomical locations of CHs with significant neural synchronization. CHs that showed significant brain-to-brain synchronization (speaker-listener) are shown in turquoise (also see arrows in turquoise between the listener and the speaker). This result corresponds to Fig. 3b. That is, based on the WTC analysis on the pairs of neural signals within each participant group, significantly enhanced INS in the LA pairs was identified at TPJ-TPJ (CH3-CH3) compared to that of the LU pairs (two-sample *t*-test, two-tailed) in both the face-to-face and back-to-back conditions. Significantly enhanced INS was also found at pSTC-TPJ in the LA pairs compared with the LU pairs in the face-to-face condition. CHs that showed significant brain-to-speech synchronization in the listener's brain are shown in yellow. CHs that showed significant brain-to-speech synchronization in the speaker's brain are shown in brown. These latter results correspond to Fig. 6, and were obtained from the calculation of neural synchronization between the brain activity and the speech envelope. These spatial patterns show the dissociation of brain-to-brain synchronization from brain-to-speech synchronization in their anatomical locations. The brain images were automatically generated using BrainNet Viewer, which is freely available on the NITRC[51]

verbal responses, and was significantly correlated with scores of communication quality. In addition, the significant enhancement of INS was also found when the listener's brain activity preceded that of the speaker, and was independent of brain-to-speech synchronization. Finally, the above findings were consistent regardless whether the participants were face-to-face or back-to-back, although visual inputs were associated with a separate increase of INS. These results are discussed below.

First, for over half a century, the neural mechanism for the cocktail-party effect has been a mystery. It is not well-understood how the brain of the listener selectively processes the attended information while discounting the other information. Previous theories have suggested that neural synchronization in different scales plays a critical role in this cognitive process[33–35]. However, little evidence exists about how it works in a multi-speaker situation. While previous research has focused on the acoustic features and their extraction within a single brain (brain-to-speech), the present research aimed to clarify the role of INS (brain-to-brain) in the cocktail-party effect. The new finding of the current study was that INS between the listener and attended speaker was selectively enhanced in a naturalistic multi-speaker situation. In addition, results from both the LA vs. LU comparisons and the FLDA suggested that the identified INS increase played an important role in the multi-speaker situation. Thus, the present findings provided supportive evidence for the perspective that neural synchronization is a potential neural mechanism for the selective processing of the attended information in a multi-speaker situation.

Second, several sets of results helped to exclude the potential confounding effect of shared sensory inputs on the increase of INS. The first set of results concerns the sensory inputs and non-communication factors. Given the design of the study, the data were collected in the same session for the LA and LU pairs, who shared the same sensory inputs (such as topics, experimental contexts, etc.), except that there were no true communications in the LU pairs. Behavioral tests further confirmed that the two speakers were comparable with regards to their speaking patterns

(speaking rate, fluency, and naturalness) and frequency of both verbal and non-verbal responses. Most importantly, the same effect was found both in the face-to-face and back-to-back conditions, suggesting that visual inputs were not a necessary condition for the selective enhancement of INS in a multi-speaker situation. Finally, the random permutation test further excluded the potential confounding effect of non-communication behaviors on the increase of INS. In sum, all of these findings suggest that sensory inputs could not explain the selective enhancement of INS in a multi-speaker situation.

The second set of results to rule out the sensory input hypothesis came from the time course analysis. The INS increase peaked at a time point preceding the occurrence of verbal responses by about 5–6 s. The relevant brain area is close to left TPJ, one of the main brain areas previously found to be involved in the cocktail-party effect[9,10,36]. Left TPJ also plays a major role in the representation of someone else's concept[37]. Consistent with this notion, previous evidence has demonstrated that INS for content processing differs from that for sensorimotor processing in terms of both anatomical location and temporal pattern. Specifically, for the anatomical location, INS that is associated with sensorimotor events is found at primary sensorimotor areas, whereas INS that is associated with content information is found at associative brain areas such as superior temporal cortex, TPJ, precuneus, and frontal cortices[13,14,18,38–40]. For the temporal pattern, the former is temporally aligned to the sensorimotor events such as speech utterance or hand movement, whereas the latter involves a time-lag between the speaker's brain activity and that of the listener[14,18,27]. Our results that the peak of INS increases was significantly correlated with quality of communication, but not with speaking patterns such as speaking rate and level of disfluency, also confirmed the association between the INS increase at TPJ and the representation of content information about the target speaker (i.e., high-level mentalizing). It was also consistent with a previous finding that during a conversation, predictability effects were present when the listeners had to prepare a verbal response, but not when they had to predict the turn-end[17]. Taken together, the peak of the INS increase at left TPJ was

likely associated with high-level content information rather than sensory-level information.

The third set of results was that the highest level of the INS increase occurred when the brain activity of the listener preceded that of the speaker by 1–3 s when the actual speech had not been produced by the speaker. Redundancy of the speech signal could not have explained this finding because no research has shown that speech redundancy leads to INS when the listener's brain activity precedes that of the speaker. Furthermore, the timescale for the processing of high-level linguistic structures such as sentences and paragraphs is seconds, whereas that of sound-level acoustic features is milliseconds[41]. Therefore, our finding of 1–3 s time-lag is more likely to reflect sentence- or even paragraph-level rather than sound-level processing.

The fourth set of results was that brain-to-speech synchronization was found to be disassociated with brain-to-brain synchronization in both the anatomical location and frequency range of the brain activity. Previous studies that employed either MEG or EEG technique have shown that the auditory cortical activity can selectively track the temporal feature of the attended speech while ignoring the unattended one. But this effect has been found to be independent of brain-to-brain synchronization in a clear auditory situation[15]. To test their relationship in a noisy situation, the present study replicated this effect in the sensorimotor and auditory cortices while using the slow fNIRS signal. We also demonstrated such an effect between fNIRS signal at IFC of the speaker and the amplitude of envelope of her/his own speech. However, neither the anatomical location nor the frequency range (around 0.8 Hz) of brain-to-speech synchronization overlapped with that of brain-to-brain synchronization (TPJ at CH3, 0.1–0.4 Hz). The potential influence of these CHs that showed significant brain-to-speech synchronization on the selective enhancement of INS at TPJ was also excluded through a covariate analysis.

The final relevant result came from a comparison between two brain areas. It is known that while both TPJ and PFC are involved in communications, they are associated with different cognitive functions, i.e., PFC is more closely associated with attention, planning, and emotion[25,26], and TPJ is more closely associated with the prediction of each other's subsequent actions[24]. In this study, we found relatively higher-level INS increase at pSTC-PFC in both the LA and LU pairs in both the face-to-face and back-to-back conditions (not significant after FDR correction). In the single-speaker tasks, the INS increase was also identified (significant at $P < 0.05$ level after FDR correction) in the LA pairs at pSTC-PFC (CH7-CH11) in both the face-to-face and back-to-back conditions. Moreover, these results were obtained when no time-lags of the brain activities were added to the data. However, we did not find a significant difference between the LA and LU pairs at pSTC-PFC probably because this type of INS was irrelevant to the cocktail-party effect. Thus, pSTC-PFC was selected as a control in the time course analysis, which showed a distinct pattern compared with TPJ-TPJ. These patterns further suggested that the INS increase at TPJ-TPJ was likely to be associated with selectively high-level content processing in a multi-speaker situation rather than with general predictions in a single-speaker situation.

## Methods

**Participants.** Sixty-six adults participated in this study. They were pseudorandomly split into 22, 3-person groups. For each group, the members had to be of the same sex (to avoid a potential confound of inter-gender interactions) and were total strangers to one another. There were 11 female groups and 11 male groups. One female group was excluded because of data collection failure. The mean age of the remaining 21 groups was 23 years (standard deviation (SD) = 2). All participants were right-handed[42] and had normal hearing. They did not have any language, neurological, or psychiatric disorders.

Written informed consent was obtained from all participants. The study protocol was approved by the Institutional Review Board of the State Key Laboratory of Cognitive Neuroscience and Learning, Beijing Normal University.

**Tasks and procedures.** For each group, an initial resting-state session of 5 min served as a baseline. During this session, the participants were required to keep still with their eyes closed, relax their mind, and remain as motionless as possible[13].

The task sessions immediately followed the resting-state session. For each participant group, one of the three participants was randomly assigned as the listener (mean age = 22 years, SD = 3), whereas the other two participants were assigned as the speakers (left speaker, mean age = 23 years, SD = 2; right speaker, mean age = 23 years, SD = 2). There were no significant differences across the three participants in age (ANOVA, $F(2, 62) = 1.639$, $P = 0.203$). Each participant group performed six tasks: two experimental conditions by three communication tasks. Due to the low inter-judge reliability in determining the target speaker in a multi-speaker task with freely chosen target (see Supplementary Methods), this task was excluded from this study, leaving two experimental conditions by two communication tasks. Six topics were carefully selected and their characteristics were matched (see below). All six topics were used in each task, and the sequence of the topics was counterbalanced across the participant groups within each task.

The two experimental conditions were face-to-face and back-to-back conditions. In the face-to-face condition (Fig. 1a), the three participants sat in a circle in a face-to-face manner. The locations of the left and right speakers were −30° (left) and +30° (right), respectively, relative to the listener (Fig. 1a). The two tasks were as follows. For the first task, one speaker talked to the listener, while the other speaker was asked to keep silent. The listener was asked to attend to the first speaker and listen carefully. The listener was also allowed to respond to the first speaker verbally or non-verbally (e.g., facial expressions or gestures). For the second task, two speakers talked to the listener simultaneously about the same topic but with different contents (i.e., they decided by themselves what to say about the topic). The listener was asked to attend to the target speaker assigned a priori while ignoring the other speaker. The listener was also allowed to respond. The same procedures were conducted in the back-to-back condition except that the participants sat in a back-to-back manner and they could not see one another. Each task lasted 5 min. Two additional 30-s resting-state periods (one at the initial phase and the other at the ending phase of each task) were used to allow the imaging instrument to reach a steady state. The sequence of the two tasks within each condition and the sequence of the two conditions were counterbalanced across the participant groups. The overall procedures were video recorded.

INS was computed between each pair of the participants. It was assumed that the comparison between the LA and LU pairs in their INS increases would reveal whether INS was involved in the selective process in a multi-speaker situation.

**Topics used in the tasks.** Six topics were selected from 12 candidate topics. The selection of the topics was based on their high levels in five aspects (difficulty, familiarity, abstractness, frequency of occurrence in daily life, and narratability) on a five-point scale (1 = low, 5 = high; difficulty was reverse-coded). An independent sample of 20 participants (male: 11; mean age = 22 years) made the judgment. The inter-judge reliability (ICC) was satisfactory to high (ranging from 0.747 to 0.894) for all aspects for all topics. The overall internal consistency across the five aspects was also high (Cronbach alpha = 0.956). Consequently, the ratings of these five aspects were averaged, and the highest rated six topics were used in the current study (mean scores = 4.31). Independent-samples Kruskal–Wallis test did not show any significant differences among the six topics with respects to difficulty ($\chi^2 = 4.055$, df = 5, $P = 0.542$), familiarity ($\chi^2 = 0.89$, df = 5, $P = 0.971$), abstractness ($\chi^2 = 5.562$, df = 5, $P = 0.351$), frequency of use in daily life ($\chi^2 = 10.017$, df = 5, $P = 0.075$), and narratability ($\chi^2 = 4.950$, df = 5, $P = 0.422$). Therefore, these topics were considered as equivalent when they were used for different conditions and tasks (face-to-face vs. back-to-back; multi-speaker vs. single-speaker). For each task, the same topic and relevant background material were provided to the two speakers in order to control for the levels of familiarity and difficulty. In addition, although the attended and unattended speakers spoke about the same topic, the contents and specific sentences were different (the speakers had time to think about what they would say after getting the topic and relevant background materials).

**Assessment of communication quality.** Immediately after each task, the participants who were involved in the task were asked to assess the quality of their communication on a five-point scale (1 = low, 5 = high). The assessment included two items: (1) How well did you understand what your partner was talking about? (2) How well did you think your partner understood what you were talking about? The assessment scores from the listener and the attended speaker were averaged to have an overall score of communication quality.

**Coding of communication behaviors.** In the multi-speaker tasks, verbal (i.e., turn-takings and interjections) and non-verbal responses (i.e., orofacial movements, facial expressions, and body gestures) were coded by two coders. The inter-coder reliability was computed at the time-point level for each individual group. The ICC reached a high level (verbal responses vs. no response, from 0.804 to 0.945 in the

face-to-face condition, and from 0.754 to 0.945 in the back-to-back condition; nonverbal responses vs. no response, from 0.771 to 0.887 in the face-to-face condition, and from 0.726 to 0.888 in the back-to-back condition). The group-level ICC for number of responses was also high (verbal responses, 0.985 in the face-to-face condition and 0.862 in the back-to-back condition; nonverbal responses, 0.844 in the face-to-face condition and 0.828 in the back-to-back condition). For time-points where there was not an agreement between coders, the two coders would discuss together until an agreement was obtained. The final analysis was conducted on the data based on the two coders' consensus.

**FNIRS data acquisition.** During the experiment, the participants sat in a quiet room. An ETG-4000 optical topography system (Hitachi Medical Company) was used to collect imaging data from the three participants of each group simultaneously. Three sets of the same customized optode probes were used. Each set had two groups of probes. The first group of probes was placed on the left hemisphere so as to cover left inferior frontal, temporal, and parietal cortices (Fig. 1b). The probes consisted of ten measurement channels (four emitters and four detectors, 30 mm optode separation). CH9 was placed just at T3 in accordance with the international 10–20 system (Fig. 1b). The second group of probes (one emitter and one detector, one measurement channel) was placed on the left dorsal lateral PFC. All probe sets were examined and adjusted to ensure consistency of the positions among the participants of each group and across the groups. Only the left hemisphere was covered because the selective attention-related scalp-positive potentials evoked by simultaneous natural speech streams are lateralized to the left hemisphere[43]. The absorption of near-infrared light at two wavelengths (695 and 830 nm) was measured with a sampling rate of 10 Hz. The changes in the oxy-hemoglobin (HbO) and deoxy-hemoglobin (HbR) concentrations were recorded in each channel based on the modified Beer–Lambert law. This study focused only on the changes in the HbO concentration, which was demonstrated to be the most sensitive indicator of changes in the regional cerebral blood flow in fNIRS measurements[44].

**Behavioral data analysis.** To ensure the comparability of the verbal and non-verbal inputs between the LA and LU pairs, patterns of communication behaviors from the two speakers that were assessed by the listener were compared using Mann–Whitney U test. Meanwhile, the calculated speaking rate and percentage of disfluency based on the experimental video were also tested by independent two-sample t-tests (two-tailed).

**Selective enhancement of INS in a multi-speaker situation.** FNIRS data collected during the resting-state and task sessions were analyzed. During pre-processing, data in the initial and ending periods (30 s resting-state period plus 10 s communication tasks, respectively) were removed, leaving 280 s of data for each session. Wavelet transform coherence (WTC) was used to assess the cross-correlation between two fNIRS time series generated by pairs of participants as a function of frequency and time. The wavelet coherence MATLAB package was used[45] (for more thorough information, please see Grinsted et al.[45] and Chang and Glover[46]). Briefly, three HbO time series were obtained simultaneously for each CH from the three participants of each group. WTC was applied to each pair of the time series to generate 2D coherence maps. Because there were 11 measurement channels for each participant, an $11 \times 11$ matrix was generated for each pair of the participant groups. According to previous studies[13,21], the coherence value should increase when there are interactions between persons relative to the resting state. Based on the same rationale, the average coherence value between 0.1 and 0.4 Hz was calculated. This frequency band excluded physiological noises such as those associated with cardiac pulsation (about 0.8–2.5 Hz) and high-frequency head movements. Finally, the coherence value was time-averaged and converted into Fisher z-value. The same procedure was applied to all tasks and the resting-state condition. It should be noted that before the WTC analysis, no filtering or detrending procedures were applied[21]. These procedures were conducted on the coherence value as described above. In addition, we also did not perform artifact corrections on the signal of individual subject, as WTC normalizes the amplitude of the signal according to each time window and thus is not vulnerable to the transient spikes induced by movements[47].

The averaged coherence value in the resting-state session was subtracted from that of the task sessions, and the difference was used as an index of the INS increase for each pair of the participants. Three-way repeated measures ANOVAs, i.e., 2 (face-to-face vs. back-to-back) $\times$ 2 (single-speaker task vs. multiple-speaker task) $\times$ 2 (LA vs. LU) were conducted with FDR correction for all CH combinations ($P < 0.05$). Significant results were followed by pairwise comparisons using two-sample t-test with FDR correction for all CH combinations ($P < 0.05$, two-tailed).

To verify that the INS increase was specific to the LA pairs, a validation approach was applied to the tasks involving competing speech. That is, for each task, all 63 participants were randomly assigned to 21 new three-member groups, and then the INS analysis was re-conducted. This permutation test was conducted 1000 times for each task.

**Prediction of the attended speaker based on the increase of INS.** FLDA was conducted to predict the attended speaker and unattended speaker. Specifically, the

INS increases at all CH combinations that were averaged across the whole time courses were used as classification features, and the pairs of the LA and LU were used as class labels. A leave-one-out cross-validation approach was employed. This analysis was conducted across all tasks.

**Level of information associated with the increase of INS.** Communication behaviors were time stamped and linked to the time course of the INS increase at CH combinations of interest. First, the time courses of the INS increase at the CH combinations of interest were down-sampled to 1 Hz to obtain point-to-frame correspondence between the signal's time course and experimental video recordings. Second, the time points of the video were marked as having either verbal or nonverbal or no responses. Third, the INS increases that ranged from $-10$ s (before response) to $+10$ s (after response) were selected as epochs, and averaged across the same category of epochs to obtain two indexes: i.e., the INS increase of verbal responses and that of non-verbal responses. Meanwhile, the INS increases corresponding to time points where no response was made (i.e., one person was talking) were also averaged to obtain an index of no responses. The INS data were adjusted for the delay-to-peak effect in the fNIRS signal (about 6 s). Finally, the INS increase of verbal and non-verbal responses were compared with that of no responses in a point-by-point manner using paired two-sample t-tests. The results were thresholded at $P < 0.05$ level (FDR correction). Moreover, the INS increase within the epoch (i.e., $-10$ to $+10$) were further correlated with overall scores of communication quality and various measurements of speech production pattern in the two speakers using Pearson correlation. In addition, a CH combination at pSTC-PFC was selected as a control.

To further test the relationship between the INS increase and the level of information processing, the INS increase was recalculated by shifting the time course of the listener's brain activity forward relative to that of the speaker's brain activity by 1–10 s (step = 1 s) and vice versa. The cross-correlation analyses and statistical tests were conducted again on each time lag.

Finally, we directly examined whether brain-to-speech synchronization contributed to brain-to-brain synchronization. To calculate the synchronization between brain activity and the speech envelope, the following steps were conducted. The experimental videos were converted into audios first. Because one of the digital video (DV) recorders was connected to the ETG-4000 equipment, it was able to be synchronized with the fNIRS data acquisition. By examining the video content, the exact starting and ending time of the task was also marked on the other DV recorder. Thus, the converted audios were exactly synchronized with the fNIRS data. Next, a blind source separation (BSS) procedure[48] was conducted using an independent component analysis (ICA) program of the EEGLAB software (version 14)[49,50]. Briefly, in a multi-speaker situation, if each of the M speeches were recorded by N microphones, the recordings will be a matrix composed of a set of N vectors, each of which is a (weighted) linear superposition of the M speeches. BSS enables extraction of the target speech from observed mixed speech without the need for source positioning, spectral construction, or a mixing system. One of the key methods for performing BSS is ICA, where we can take advantage of (assumed) linear independence between the sources. In this study, two DV-linked microphones were placed in two separate positions to record the speeches of each speaker (Fig. 1a). Based on this, the ICA program produced two ICs for each participant group. The ICs were labeled and saved as attended and unattended speeches, respectively, according to the cross-checking of the contents between the ICs and the video. For each of the separated speeches, the amplitude envelope was computed using the envelope function of MATLAB and resampled to 10 Hz (the same as the fNIRS data). Then, WTC was applied to compute the coherence values between the listener's fNIRS signal and the attended and unattended speeches, and between the two speakers' fNIRS signal and their speeches, in the multi-speaker task in both the face-to-face and back-to-back conditions. Meanwhile, this computation was also conducted between the fNIRS signal at the resting-state and the speeches in the tasks, which were used as control data as no brain-to-speech synchronization was expected. Finally, paired two-sample t-test (two-tailed) was conducted to examine whether there was significant brain-to-speech synchronization in the multi-speaker tasks compared with the control data, and whether brain-to-speech synchronization, if there was any, differed significantly between the LA and LU pairs along the full frequency range (0.01–0.8 Hz). Data above 0.8 Hz were not considered according to previous BOLD evidence[28,29]. Because there were no significant results after FDR correction across all measurement CH-speech pairs over the full frequency ranges (11 pairs in total, 0.01–0.8 Hz) at $P < 0.05$ level, uncorrected results at a significance level of $P < 0.0005$ were reported. To further exclude the potential influence of the CHs' brain-to-speech synchronization on brain-to-brain synchronization, the INS increases for CH combinations related to brain-to-speech synchronization were averaged and used as the covariates when comparing the INS increase between the LA and LU pairs using ANCOVA.

**Code availability.** All analyses were performed using Matlab R2016b, with standard functions and toolboxes. All code is available upon request.

**Data availability.** The data that support the findings of this study are available from the corresponding author upon reasonable request.

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

## Acknowledgements

This work was supported by National Natural Science Foundation of China (31622030 and 31411130158), the Fundamental Research Funds for the Central Universities (2017EYT32 and 2017XTCX04), and the Open Research Fund of the State Key Laboratory of Cognitive Neuroscience and Learning (CNLYB1605 and CNLZD1604).

## Author contributions

B.D., C.L., and C.C. conceived the experiments. B.D., L.Z., and Y.L. performed the research. B.D., C.C., Y.L., L.Z., H.Z., X.B., W.L., Y.Z., L.L., T.G., G.D., and C.L. analyzed the data. C.L., C.C., and B.D. wrote the paper.
