## [Peer Review File · Nature Communications]

Reviewers' comments:

Reviewer #1 (Remarks to the Author):

In this paper Dai et al. use a three-person communication task and simultaneous functional Near-Infrared Spectroscopy (fNIRS) to study the neural underpinnings of the cocktail-party effect: the phenomenon whereby a person at a cocktail party is able to and hold a conversation with a person and tune out other speakers. The paradigm used is a 2x3 design. The first two factors are facing and not-facing: the three subjects either face each other or they don't. The second three factors are single, free conversation, and experimenter selected partner. In the single condition one subject, the Listener (who is fixed in that role across the 6 conditions) is paired with a designated Speaker, who engages the Listener on a set topic. The other participant remains silent. In the free conversation condition, both Speakers engage the Listener (who again is free to respond) on the same topics, and the Listener is free to attend to either (or both) speaker. All three subjects undergo fNIRS scanning during the task (a pre-task resting state scan is also used to establish a baseline) The authors compute the cross-coherence matrices (INS – interpersonal neural synchronization) of the Listener and the two Speakers for each of the cells of the design (a wavelet transform is used). Additionally, the authors look at correlations between (increases from baseline) coherence and "increase in mutual understanding" in selected ("ROI" from the cross-coherence analysis). For the free-conversation condition an "attended Speaker" was selected by two "independent coders." The authors compute a one-sample t-test on the cross-coherence maps for each of the three pairings (LA,LU,AU) for each of the 2 x 3 (2?) conditions. Channel coherences that were significant at this stage (?) were further subjected to comparison across the conditions LA-LU, LA-AU, LU-AU.

The central results are that 1) for the assigned condition (TPJ) for CH3xCH3 the differences LA-LU, LA-AU were significant, while LU-AU were not; 2) for the free condition the comparison LA-LU for CH3x? was significant, but not LA-AU or LU-AU; 3) for the single Speaker condition, results referring to significant increases in coherence within a condition are stated, but not across-pairing comparisons are noted.

General Impression.

This is an interesting paradigm, the use of hyper-scanning is quite novel, and consequently there is a voluminous amount of data to communicate.

Major Issues.

1. The central results are somewhat difficult to follow. Specifically, the results pertain to entries in a matrix, a coherence matrix, for example the matrix of coherences for LA, or LU. When results about these entries are discussed both indices (one for the L, one for the, for example, A). For example this notation might work: for the 3,7 entry in the LA coherence matrix (incidentally, L-A should not be the notation because subtractions across conditions are used and L-A is not a subtraction). This could go a ways to cleaning up and streamlining the exposition.

2. It seems according to the methods that the comparisons across speakers – for example, in the assigned condition – are only computed for "areas" (pairs of indices i,j) that are significant at the individual condition level. This is not statistically justified (cf. Kreigskorte, Nature Neuroscience). The question "which i,j's in condition A are statistically different than (bigger than) entries i,j in condition B?" can be answered by a two-sample t-test across all the conditions with the threshold adjusted for all of the multiple comparisons.

3. The results comparing LA with say LU within a condition are quite interesting, but isn't the comparison LA in the free conversation condition, versus LA in the single condition, also relevant to the question? Or the same question across free/chosen? It is very interesting to see the differences between the coherences LA and LU, but the across condition comparisons get to the issue of what is really unique about the free, natural conversation (the cocktail party).

4. This isn't really an issue, but rather an observation or suggestion. Have the authors considered a prediction approach in the free conversation condition? Can the fNIRS measurements (the 11x11 coherence matrix) be used to predict which pair is being examined (LA v LU, for example). This also raises a question. The "coders" were used to categorize the speakers as either the attended or unattended speaker. Could a value in [0,1] be calculated instead and be used in a predictive framework, or otherwise?

Minor Issues.

1. CH clearly stands for channel, but it is never defined. All abbreviations should be defined.

2. Figure 1 is not particularly edifying. It and the model don't add very much to the ms.

3. Figure 2A legend line 634: "placed in opposite directions" is not very informative. Further, Figure 2A is difficult to decipher. It might be better to stick to Figure 2B&C and reserve A for the supplementary materials.

4. Line 209: "in a competing situation" is ambiguous.

5. Line 395: why are the t-maps smoothed? Were smoothed t-values used in the analysis or the raw t-values?

6. Figure 4A/2 is very difficult to understand – it doesn't add much. Supplementary material?

7. Figure 4B&E – it is difficult to understand from the methods how these panels were made.

8. Line 424: "The INS increases in the COI ..." the statistics appear not to be described.

Reviewer #2 (Remarks to the Author):

This novel study shows that interpersonal neural synchronization (INS) in the left TPJ is enhanced between a listener and an attended speaker compared to an unattended speaker. The authors also report interesting correlations between this synchronization and behavior. This work bridges literature on the cocktail party effect and neural synchronization and is a nice addition to both fields.

However, I have several reservations which prevent me from recommending acceptance in the paper's current form.

1. Methodological terms ("INS", "communication quality") should be defined in brief when introducing them in the Results to give readers a sense of the analysis procedures before the Methods.

2. Were any procedures taken to ensure that motion artifacts did not interfere with the data? How was co-registration performed across subjects (as the precise location of specific channels can vary)?

3. The authors should discuss what it means for “rest” to be a baseline. During rest, you’d expect to see uncoupling of individuals because people should be thinking about varied topics. Why not have them to do a similar, non-communication (e.g., motor or visual) task? I’d like to see the INS matrices for the baseline period.

4. I wonder why the speakers were asked to speak about same topic; this is never explained. I’m concerned that this might have induced attentional interference and unintentional switching between the two speakers. To measure this, the authors could have tested memory for the attended and unattended streams.

5. I would like more details on the tendencies of different listeners to respond (verbally or gesturally) to speakers. Did the amount or type of responses relate to any other measures?

6. I’m confused how “mutual understanding” is correlated with INS over time (Fig. 4B). Was mutual understanding a scalar value for the entire task, and INS at each time point was correlated with that value across subjects? This choice needs to be justified more clearly. A better way to do this would be to have some continuous measure of understanding (e.g., ask people to transcribe what they heard after the task or continuously report understanding during the task) and correlate this throughout the task, since presumably mutual understanding varied over time.

7. It’s interesting that INS was best correlated with “mutual understanding” prior to a verbal response. The authors should cite and relate their findings to several studies (Liu et al, Scientific Reports, 2017; Stephens et al, 2010) that found strongest neural coupling at a lag in the time series of ~6-10 seconds (meaning that speakers’ activity preceded listeners’ activity by a few seconds, due to anticipatory preparation of upcoming speech). How does this finding relate to this previous work (and can the authors replicate the lag correlation result with their data)? I’m also surprised the authors don’t expect PFC to be involved in this process and even use it as a control region here. Please explain this discrepancy.

8. Figure 1 is very poorly organized; the flow between graphics is not clear, and it should be completely re-vamped.

9. Plot labels should be clearer (e.g., show “listener”/“speaker” on matrix axes)

Reviewer #3 (Remarks to the Author):

In this paper the authors use fNIRS technology in order to measure neural activity simultaneously from 3 participants engaged in a “social context”, and attempt to quantify the inter-personal neural synchronization among them as a function of attention and communication quality. I find the design and experiment to be novel and timely, as it aims to address an issue that has long interested the cognitive neuroscience community regarding inter-personal communication.

This is also, however, a methodologically challenging feat, and requires many controls, which I am not sure are all properly addressed in the current manuscript.

My first concern is that when two people are engaged in a conversation, they share many common sensory inputs (primarily auditory) which would manifest in similar neural responses that also termed INS but would actually be trivial. The authors do not address this head-on, and it is a critical point before making claims about effects of attention and higher-order cognitive functions needed for communication.

My second main concern is that monitoring neural activity in this type of naturalistic experiment makes it difficult to relate specific neural response to behavior, as it is not controlled. While the authors clearly made an effort to code some aspects of the behavior diligently, clearly not all "talking" responses can be considered the same. It is critical to provide detailed information about the statistics of the behaviors monitored (see specific comments below).

My third main concern regards the comparison of face-to-face vs. back-to-back conditions. Clearly engaging in a face-to-face interaction, which involves multiple additional visual and motor cues is expected to increase INS as well as the efficacy behavioral communication. Yet in the presented data there seems to be no differences. This comparison should be more broadly explored.

Specific Comments:

Introduction (and abstract) should contain a more detailed explanation of what INS is - e.g. what aspects of neural activity are synchronized? How is it measured?

Methodological questions:

Where were the participants looking in the face-to-face condition? If the listener was looking at the attended speaker, this will obviously affect their neural synchronization (as well as speech-tracking responses in auditory cortex and other brain regions), and the comparison with the unattended speaker is no longer valid/comparable.

Were the participants given instructions about what to talk about? (e.g. what were the topics?) Also, did the two "talkers" know which one was attended? Is it possible that if you know you are designated as an "attended" (and engage in a conversation) or "unattended" (speaking to yourself), this will change the way you talk (speed, content, enthusiasm etc.)

In the condition where the attended speaker was "freely chosen" by the listener - how did the coders determine who was attended in the back-to-back condition? Also, did the listener tend to select one speaker in this condition or to "jump" between them? And were the communication ratings and frequency of response different in this condition? Please provide some behavioral data on this.

Also, please provide behavioral data on the face-to-face vs. back-to-back conditions

Could both speakers hear the acoustic environment (i.e. what they and the other person were saying?)

The Verbal behavioral measures coded indicate that the listener also spoke during the experiment (a point not mentioned when describing the experiment). Could you please add information regarding the communication and speech dynamics - what % of the time did each speaker speak, what was the dynamic of turn-taking/interjections, can you assess the speech fluency of each speaker (rate of speech, pauses etc); and conversely for the unattended speaker - did she have more pauses/less fluent speech than the attended+listener

Also, perhaps it is incorrect to refer to the participants as: listener, attended and unattended, but rather as one communicating pair and another individual talking to herself?

Was the analysis focused only on the parts of the session where she was silent?

Results

Is it possible to tease apart the aspects of the INS measure which pertain to encoding the acoustic input (which obviously both the listener and the speakers share; and therefore might be trivial), and aspect that are more selective and related to attention? The permutation test which mis-matched listeners and speakers from different sessions does not account to the inherently shared acoustic within-session...

Although the statistics were done on individual sensor-pairs, and few pairs passed the statistical threshold, the correlation matrices shown in Figure 3 suggest that the INS between L-attended is not

really specific to one "pair" of sensors, but is actually broader and found in all condition. E.g., if you were to compare the mean INS in all pairs across conditions, it seems likely you would find patterns not represented in the data as is. Perhaps a multi-voxel pattern analysis would be more appropriate, and/or aggregate statistics, rather than multiple individual t-test + corrections, which by definition reduce your ability to discover real effects.

Minor

Abstract "while they communicated naturalistically" - should be rephrased

Also, please refrain throughout the manuscript from referring to "the cocktail party effect" as a measure/manipulation, e.g. "the cocktail party effect was localized to TPJ" (line 163); "we expected a greater INS increase for the cocktail party effect" (p. 5)

Reviewer #4 (Remarks to the Author):

The paper "selective enhancement of interpersonal neural synchronization underlies the cocktail party effect" studies the synchronization between the neural signals in the brains of a listener and the attended and unattended speakers using fNIRS signals.

The authors show increased interpersonal neural synchrony (INS) in left TPJ between the listener and attended speaker which also correlates with the quality of listening. The authors suggest that INS may be the neural precursor of cocktail party effect.

The questions asked in the study are important, the experiments are well designed, and the analysis of the neural data is sound. However, I am quite uncertain about the implication of the findings and the claims that are made regarding "whether and how the INS underlie the extraction of the target speech". Specifically:

- In general, it has been shown that the acoustic waveform that is produced by a speaker is highly correlated with the movement of the articulators (lips, jaw, tongue, ...), and the movement of the articulators is highly correlated with the neural activity in the speaker (e.g. Bouchard et. al. Nature 2013). The acoustic signal is also highly correlated with the neural activity in the brain of listener (Ding et. al., Zion-Gollumbic, et. al., ...), and many studies have shown that the neural signals in a listener track only the attended speaker (Ding et. al, Zion-Gollumbic et.al, O'Sullivan et. al. among many more).

Given all these, I am not sure why the findings of this study are novel or surprising, where what is shown is basically the tracking of the attended speaker in the listener's brain. In other words, this could be explained simply by the auditory manifestation of speaker's articulatory movements, and its encoding in the brain of a listener.

The only argument that the authors present in the paper against the hypothesis that the increased INS reflects merely auditory tracking is the delayed verbal response which comes later than the INS increase. However, the authors have no way of knowing when the subjects actually start tracking the attended speaker. It well may be that the listener attends and tracks a speaker for few seconds before giving a verbal or non-verbal response, therefore, it cannot be taken to indicate the lack of auditory tracking prior to the response.

- The authors claim that their study supports the perspective that neural synchrony is a general mechanism underlying selective grouping of object features in a competing situation. The temporal coherence as a binding mechanism of acoustic features (which I assume is the one the authors are

referring to) is very different from what the authors show in this study (< 0.4 Hz), it is very unclear how their finding provides supporting evidence for this mechanism.

- "The peak of the INS increase in the left TPJ likely marks the achievement of mutual understanding in the cocktail party effect." I am not quite sure what they mean by mutual understanding when only one person talks the whole time (speaker vs. listener).

- I also find statements like "INS increase in TPJ played a specific role in selectively extracting the target speech" not compelling since the authors haven't actually shown a role for TPJ in extracting the target speech (only an increase in INS measure, which doesn't necessarily mean a causal mechanism).

- Increased TPJ during freely selecting target compared to fixed target is interesting, however, it is not clear what the authors mean by "listener had to hold all information". Is it possible to actually show this? For example, do they perceive the unattended better in this situation compared to assigned target?

One point that casts doubt on this finding is the way the authors "guess" the attended speaker. One may argue that inaccuracies in specifying the actual attended speaker may partly explain this difference, since the metrics the authors used are not based on comprehension, but based on behavioral indicators such as the direction of the head or nodding patterns which inherently is not very accurate.

Minor: Arnott et. al. 2004 reference is missing.

Response to reviewers' comments

Reviewers' comments:

Reviewer #1 (Remarks to the Author):

In this paper Dai et al. use a three-person communication task and simultaneous functional Near-Infrared Spectroscopy (fNIRS) to study the neural underpinnings of the cocktail-party effect: the phenomenon whereby a person at a cocktail party is able to and hold a conversation with a person and tune out other speakers. The paradigm used is a 2x3 design. The first two factors are facing and not-facing: the three subjects either face each other or they don't. The second three factors are single, free conversation, and experimenter selected partner. In the single condition one subject, the Listener (who is fixed in that role across the 6 conditions) is paired with a designated Speaker, who engages the Listener on a set topic. The other participant remains silent. In the free conversation condition, both Speakers engage the Listener (who again is free to respond) on the same topics, and the Listener is free to attend to either (or both) speaker. All three subjects undergo fNIRS scanning during the task (a pre-task resting state scan is also used to establish a baseline) The authors compute the cross-coherence matrices (INS – interpersonal neural synchronization) of the Listener and the two Speakers for each of the cells of the design (a wavelet transform is used). Additionally, the authors look at correlations between (increases from baseline) coherence and “increase in mutual understanding” in selected (“ROI” from the cross-coherence analysis). For the free-conversation condition an “attended Speaker” was selected by two “independent coders.” The authors compute a one-sample t-test on the cross-coherence maps for each of the three pairings (LA,LU,AU) for each of the 2 x 3 (2?) conditions. Channel coherences that were significant at this stage (?) were further subjected to comparison across the conditions LA-LU, LA-AU, LU-AU.

The central results are that 1) for the assigned condition (TPJ) for CH3xCH3 the differences LA-LU, LA-AU were significant, while LU-AU were not; 2) for the free condition the comparison LA-LU for CH3x? was significant, but not LA-AU or LU-AU; 3) for the single Speaker condition, results referring to significant increases in coherence within a condition are stated, but not across-pairing comparisons are noted.

General Impression.

This is an interesting paradigm, the use of hyper-scanning is quite novel, and consequently there is a voluminous amount of data to communicate.

Major Issues.

1. The central results are somewhat difficult to follow. Specifically, the results pertain to entries in a matrix, a coherence matrix, for example the matrix of coherences for LA, or LU. When results about these entries are discussed both indices (one for the L, one for the, for example, A). For example this notation might work: for the 3,7 entry in the LA coherence matrix (incidentally, L-A should not be the notation because subtractions across conditions are used and L-A is not a subtraction). This could go a ways to cleaning up and streamlining the exposition.

Answer1 (A1): Following this reviewer's helpful suggestion, we have changed the notations of the L-Attended and L-Unattended pairs to the LA and LU pairs, respectively, to avoid potential confusion. In addition, we have clarified the presentation of the matrices of coherence.

2. It seems according to the methods that the comparisons across speakers – for example, in the assigned condition – are only computed for “areas” (pairs of indices i,j) that are significant at the individual condition level. This is not statistically justified (cf. Kreigskorte, Nature Neuroscience). The question “which i,j 's in condition A are statistically different than (bigger than) entries i,j in condition B?” can be answered by a two-sample t-test across all the conditions with the threshold adjusted for all of the multiple comparisons.

A2: Thanks for the suggestion. We have recalculated the differences between the INS increase of the listener and attended speaker (LA) pairs and that of the listener and unattended speaker (LU) pairs across all measurement channels using two-sample *t*-tests. In tasks with *a priori* assigned target speaker, results still showed significant differences between the LA and LU pairs in the INS increases at left TPJ-TPJ [CH3-3, face-to-face: $t(20) = 4.049$, $P < 0.001$; back-to-back: $t(20) = 3.756$, $P < 0.001$] and pSTC-TPJ [CH7-3, face-to-face only: $t(20) = 3.878$, $P < 0.001$] after FDR correction across all measurement channels. No other CHs showed significant differences between the LA and LU pairs ($P > 0.05$, FDR correction).

In addition, in a naturalistic multi-speaker situation, the listener sometimes responded to the speaker and spoke continuously. The continuous speech occupied about 10% and 11% of the 280s on average in the face-to-face and back-to-back conditions respectively. To test the INS increase

when the listener was silent, time points when the listener spoke continuously were excluded. Again, significant differences were found between the LA and LU pairs in the INS increase at TPJ-TPJ [CH3-3, face-to-face: $t(20) = 4.126, P < 0.001$; back-to-back: $t(20) = 3.855, P < 0.001$] and pSTC-TPJ [CH7-3, face-to-face only: $t(20) = 4.056, P < 0.001$]. No other pairs of CHs showed significant differences between the LA and LU pairs ($P > 0.05$, FDR correction).

In tasks with freely chosen target speakers, the differences between the LA and LU pairs in the INS increase at TPJ-TPJ were significant in both the face-to-face [$t(20) = 3.037, P = 0.003$] and back-to-back [$t(20) = 3.887, P < 0.001$] conditions, although these effects did not survive the FDR correction across all measure channels ($P < 0.05$).

3. The results comparing LA with say LU within a condition are quite interesting, but isn't the comparison LA in the free conversation condition, versus LA in the single condition, also relevant to the question? Or the same question across free/chosen? It is very interesting to see the differences between the coherences LA and LU, but the across condition comparisons get to the issue of what is really unique about the free, natural conversation (the cocktail party).

A3: According to this suggestion, we conducted 3-way 2 (face-to-face vs. back-to-back) \times 3 (single speaker vs. assigned target vs. freely chosen target) \times 2 (LA vs. LU) repeated measure ANOVAs to see whether the increase of INS was consistent across different contexts, corrected for all CHs (FDR correction, $P < 0.05$). A significant main effect for the LA vs. LU comparison was found at TPJ-TPJ only ($F_{(1,20)} = 68.459, P < 0.001$). No other significant effect (either main effect or interaction) was found at TPJ-TPJ ($P > 0.05$). Also, no significant main effects were found for mode of communication, number of the speakers, and 2 or 3 ways interactions for other CHs ($P > 0.05$).

Although the interactions among the factors did not reach significance, we still examined the pairwise comparisons in order to confirm the two-sample t -test results. Results showed significant differences between the LA and LU pairs at TPJ-TPJ in the multi-speaker tasks but not in the single-speaker tasks, which was consistent with the results of the two-sample t -tests. Meanwhile, a significant difference between the LA and LU pairs was found at pSTC-TPJ in the single-speaker task of the face-to-face condition. The LA-LU difference at pSTC-TPJ in the multi-speaker tasks of the face-to-face condition reached significance but did not survive the FDR correction ($P < 0.05$). This result was also consistent with the two-sample t -test. No other significant results were found. These findings suggest that the INS increase is a general mechanism for social communications. However, this conclusion might be limited by the univariate statistical tests. Thus, we additionally conducted multivariate statistical tests to examine the patterns of the INS increase rather than that at

a specific brain area (see our response to this reviewer’s comment #4, i.e., A4). The multivariate statistical tests suggested a specific role of INS increase at TPJ-TPJ in a multi-speaker situation.

4. This isn’t really an issue, but rather an observation or suggestion. Have the authors considered a prediction approach in the free conversation condition? Can the fNIRS measurements (the 11x11 coherence matrix) be used to predict which pair is being examined (LA v LU, for example). This also raises a question. The “coders” were used to categorize the speakers as either the attended or unattended speaker. Could a value in [0,1] be calculated instead and be used in a predictive framework, or otherwise?

A4: Following this reviewer’s excellent suggestion, we additionally conducted a series of Fisher Linear Discriminate Analyses (FLDA). First, FLDA was applied to the multi-speaker tasks with chosen target. Specifically, INS increases at all CHs that were averaged across the time course were used as classification features, whereas the LA and LU pairs were used as class labels. A leave-one-out cross-validation approach was employed. Results showed that 85% of the LA and 88% of the LU pairs were correctly predicted (total accuracy = 87%). The same procedures were applied to the other tasks. The averaged accuracy reached 82% for the LA and 88% for the LU (total accuracy = 85%) pairs across all tasks. Table 1 summarizes these results.

Table 1 Prediction accuracy based on the increase of INS.

Conditions	Tasks	LA (%)	LU (%)	Total (%)
Face-to-face	Single speaker	95	95	95
	Multi-speaker with assigned target	76	91	83
	Multi-speaker with freely chosen target	85	88	87
Back-to-back	Single speaker	91	91	91
	Multi-speaker with assigned target	71	86	79
	Multi-speaker with freely chosen target	77	77	77

In each task, a different set of CHs showed significant contributions to the prediction, suggesting that FLDA is a more sensitive method than univariate comparison in reflecting the differences among tasks. However, only the CH at TPJ was shared as a significant predictor among all the multi-speaker tasks, but not for the single speaker tasks.

In addition, to further confirm that the INS increase is a neural correlate of the cocktail-party effect, we used the INS increase at each time point to predict which speaker the listener attended to in the multi-speaker tasks with freely chosen target. The averaged accuracy among 21 participant

groups was 77% for the attended target, 76% for the unattended target, and 77% in total in the face-to-face condition. In the back-to-back condition, the averaged accuracy was very low: 60% for the attended target, 58% for the unattended target, and 59% in total. This was probably due to the greater number of coding errors in the back-to-back condition than in the face-to-face condition (i.e., less information can be used to classify the attended/unattended target in the back-to-back condition). TPJ-TPJ was still among the top CHs that contributed significantly to the predictions.

In sum, these findings confirmed that TPJ played a critical role the cocktail-party effect, and suggested that other brain areas might play supporting roles. These new analyses and results have been added to the manuscript (see subhead of “Prediction of the attended speaker based on the increase of INS” under RESULTS, DISCUSSION, and METHODS).

Minor Issues.

1. CH clearly stands for channel, but it is never defined. All abbreviations should be defined.

A5: Thanks for catching this. We have now spelled it out the first time we mention “CH” (see line 105).

2. Figure 1 is not particularly edifying. It and the model don’t add very much to the ms.

A6: Figure 1 has been removed from the manuscript.

3. Figure 2A legend line 634: “placed in opposite directions” is not very informative. Further, Figure 2A is difficult to decipher. It might be better to stick to Figure 2B&C and reserve A for the supplementary materials.

A7: Figure 2A has been moved to Figure S1, and Figure 2B and 2C was changed to Figure 2A and 2B. Moreover, Figure 2A has been edited to mark the positions of the cameras, and the description in the legend has been changed to “One camera was placed to the left of Speaker 1, whereas the other camera was placed to the left of the listener.”

4. Line 209: “in a competing situation” is ambiguous.

A8: The phrase of “in a competing situation” has been changed to “in a multi-speaker situation”. This edition has been made throughout the manuscript.

5. Line 395: why are the t-maps smoothed? Were smoothed t-values used in the analysis or the raw

t-values?

A9: We are sorry for the misleading statement. The smoothing procedure was used only when a 2-D map was plotted (e.g., Figure S1A in the original manuscript) in order to have a continuous map. Actually it is a default step for the function of “imagesc” in Matlab. Thus, this statement has been removed. It should be noted that no smoothing procedures were used elsewhere.

6. Figure 4A/2 is very difficult to understand – it doesn’t add much. SUPPLEMENTARY material?

A10: Figure 4A has been moved to Figure S2.

7. Figure 4B&E – it is difficult to understand from the methods how these panels were made.

A11: We have revised METHODS to make it clearer (see lines 547-565).

8. Line 424: “The INS increases in the COI ...” the statistics appear not to be described.

A12: Pearson correlation analysis was conducted between the increase of INS (task-rest) and the scores of communication quality and between the increase of INS and communication patterns of the speakers. The statements have been revised (see lines 561-565).

Reviewer #2 (Remarks to the Author):

This novel study shows that interpersonal neural synchronization (INS) in the left TPJ is enhanced between a listener and an attended speaker compared to an unattended speaker. The authors also report interesting correlations between this synchronization and behavior. This work bridges literature on the cocktail party effect and neural synchronization and is a nice addition to both fields.

However, I have several reservations which prevent me from recommending acceptance in the paper’s current form.

1. Methodological terms (“INS”, “communication quality”) should be defined in brief when introducing them in the Results to give readers a sense of the analysis procedures before the Methods.

A1: We thank this reviewer for the helpful suggestion. We have revised the first paragraph of RESULTS by introducing more details of the data analyses (see lines 87-121).

2. Were any procedures taken to ensure that motion artifacts did not interfere with the data? How was co-registration performed across subjects (as the precise location of specific channels can vary)?

A2: First, we selected a frequency band of 0.1-0.4 Hz to calculate INS. Thus, high-frequency noise such as those resulted from motion artifacts (e.g., spikes) and low-frequency drifting of the signal such as those resulted from slow body movements have been removed.

Second, because fNIRS does not have anatomical information, we used the international 10-20 system to locate the position of each channel and to ensure consistency across participants during the experiment. In addition, the spatial resolution of fNIRS is about 2-3 cm¹, which is large enough for us (and all fNIRS researchers) to be confident that the same channels measure the same brain areas across participants.

3. The authors should discuss what it means for “rest” to be a baseline. During rest, you’d expect to see uncoupling of individuals because people should be thinking about varied topics. Why not have them to do a similar, non-communication (e.g., motor or visual) task? I’d like to see the INS matrices for the baseline period.

A3: This is an insightful comment. For the following reasons, however, we believe our current control/comparisons are preferred over a non-communication condition. Specifically, the present study used an fNIRS system with continuous wave. The disadvantage of continuous wave systems is that they cannot fully determine the optical properties of tissues (i.e. light scattering (μ_s') and absorption (μ_a) coefficients) and therefore absolute levels of the oxy-hemoglobin (HBO) and deoxy-hemoglobin (HBR) cannot be determined². Thus, fNIRS only measures relative changes rather than absolute levels of HBO and HBR concentration in the brain, and uses the relative changes of HBO and HBR as indexes for brain activity². Although we can compute the INS matrices for the baseline period (i.e., the resting state), the results cannot directly tell us anything about brain activity (i.e., the relative changes of HBO or HBR). Moreover, the INS matrices for the resting state were aliased by superficial blood flow in the skin of the head^{3,4,5}, resulting in less reliable INS estimation^{6,7}. The subtraction of INS at rest from that of task can solve these issues². Thus, we only presented the INS increase for task-rest rather than that for task or rest only. For comparisons, we used results in other control conditions such as the LA, LU, and AU (i.e., between the attended and unattended speakers) pairs, as well as those in the single-speaker tasks.

In this study, we used the resting state as a baseline in the first step in order to have an overall picture about how many brain areas were involved in the synchronization in the specific task. In the second step, we compared the INS increase between the LA and LU pairs [i.e., $(LA_{\text{task}}-LA_{\text{rest}})-(LU_{\text{task}}-LU_{\text{rest}})$], to control for factors the reviewer mentioned above such as the articulatory movements or common visual inputs. Between the LA and LU pairs, factors such as topics, experimental contexts, etc., were all the same except that there were no true communications in the LU pairs. Moreover, the data were collected in the same session for the LA and LU pairs. Thus, the LU pairs were an ideal control for this experiment rather than an independent non-communication task. Third, the single-speaker tasks were used as an additional control to identify the INS increase that was specific to the multi-speaker situations, and the back-to-back condition was used as another control to identify the INS increase that was independent of the common visual inputs between listeners and speakers. Finally, the random permutation test further excluded the potential confounding of non-communication behaviors on the increase of INS.

References:

1. Cui X, Bray S, Bryant DM, Glover GH, Reiss AL. A quantitative comparison of NIRS and fMRI across multiple cognitive tasks. *Neuroimage* **54**, 2808-2821 (2011).
2. Scholkmann F, *et al.* A review on continuous wave functional near-infrared spectroscopy and imaging instrumentation and methodology. *NeuroImage* **85, Part 1**, 6-27 (2014).
3. Gagnon L, Perdue K, Greve DN, Goldenholz D, Kaskhedikar G, Boas DA. Improved recovery of the hemodynamic response in diffuse optical imaging using short optode separations and state-space modeling. *Neuroimage* **56**, 1362-1371 (2011).
4. Gagnon L, Cooper RJ, Yucel MA, Perdue KL, Greve DN, Boas DA. Short separation channel location impacts the performance of short channel regression in NIRS. *Neuroimage* **59**, 2518-2528 (2012).
5. Gagnon L, Yucel MA, Boas DA, Cooper RJ. Further improvement in reducing superficial contamination in NIRS using double short separation measurements. *Neuroimage* **85 Pt 1**, 127-135 (2014).
6. Lu CM, Zhang YJ, Biswal BB, Zang YF, Peng DL, Zhu CZ. Use of fNIRS to assess resting state functional connectivity. *Journal of neuroscience methods* **186**, 242-249 (2010).

7. Zhang YJ, Lu CM, Biswal BB, Zang YF, Peng DL, Zhu CZ. Detecting resting-state functional connectivity in the language system using functional near-infrared spectroscopy. *Journal of biomedical optics* **15**, 047003 (2010).

4. I wonder why the speakers were asked to speak about same topic; this is never explained. I'm concerned that this might have induced attentional interference and unintentional switching between the two speakers. To measure this, the authors could have tested memory for the attended and unattended streams.

A4: We used the same topic in the LA and LU pairs in order to control for the levels of familiarity and difficulty across the topics. Similarly, previous studies also asked two speakers to read out the same materials and then the speeches of the two speakers were mixed into a single channel and delivered to both ears of the listener in a cocktail-party context^{8,9}.

In addition, although the attended and unattended speakers spoke about the same topic, the contents and specific sentences that the two speakers spoke were different (the speakers had time to think about what they would say after getting the topic and relevant background material). Thus, we believe that this design struck a balance between the need for naturalistic conversations and the need for comparable contents. We have added explanations about this issue in the revised manuscript.

As for a test of their memories, in the pilot phase of this study we actually tried to ask the listener to assess how well she/he understood (or what they remembered from) what the speakers were saying. But the listener felt too weird to assess because she/he did not pay attention to the unattended speaker. Thus, no such assessments were administered in the experiment. Of course, an objective recognition test was impossible because the subjects were having natural conversations, so we could not have known beforehand what the speakers would say in order to have a recognition task at the end of the conversations, let it alone a standardized one across subjects.

References:

8. Ding N, Simon JZ. Emergence of neural encoding of auditory objects while listening to competing speakers. *Proceedings of the National Academy of Sciences of the United States of America* **109**, 11854-11859 (2012).
9. Mesgarani N, Chang EF. Selective cortical representation of attended speaker in multi-talker speech perception. *Nature* **485**, 233-236 (2012).

5. I would like more details on the tendencies of different listeners to respond (verbally or gesturally) to speakers. Did the amount or type of responses relate to any other measures?

A5: The average frequencies of responses from the listeners are summarized in Table 2 below and provide as Table S1 in the revised manuscript. The responses included verbal responses such as turn-taking and interjections and non-verbal responses such as orofacial movements, facial expressions, and body gestures. An ANOVA on the frequency of responses showed significant main effects for mode of communications ($F_{(1,20)} = 13.227, P = 0.002$), number of speakers ($F_{(1,20)} = 5.283, P = 0.032$), and type of responses ($F_{(1,20)} = 40.386, P < 0.001$). In particular, a significant three-way interaction was found ($F_{(1,20)} = 5.955, P = 0.024$). No significant two-way interactions were found (mode of communications by number of speakers: $F_{(1,20)} = 1.305, P = 0.267$; mode of communications by type of responses: $F_{(1,20)} = 0.001, P = 0.983$; number of speakers by type of responses: $F_{(1,20)} = 1.375, P = 0.255$).

Further pairwise comparisons indicated that the frequency of non-verbal responses differed significantly between the face-to-face and back-to-back conditions in both the single-speaker and multi-speaker tasks (single-speaker: $P < 0.001$; multi-speaker with assigned target: $P = 0.02$; multi-speaker with chosen target: $P = 0.001$). Frequency of verbal responses differed significantly between the face-to-face and back-to-back conditions only in the multi-speaker tasks with freely chosen target ($P = 0.001$). These findings were consistent with the expectation that while verbal communications were shared between the face-to-face and back-to-back conditions, non-verbal communications were employed mainly in the face-to-face condition. Verbal and non-verbal communications seemed to be integrated to a higher level in the multi-speaker tasks with freely chosen target than in other tasks.

Table 2 Frequency of responses in the listener in various tasks.

		Verbal	Non-verbal
Face-to-face	Single speaker	11%	6%
	Multi-speaker with assigned target	11%	5%
	Multi-speaker with freely chosen target	8%	6%
Back-to-back	Single speaker	8%	3%
	Multi-speaker with assigned target	9%	1%
	Multi-speaker with freely chosen target	6%	1%

Finally, Pearson correlation analyses were conducted between frequency of responses and quality of communication and INS increase. However, no significant correlations were found ($P < 0.05$, Šidák correction).

Because we believe that the above results are not essential to the study, we did not include them in the manuscript. However, if the reviewer sees value of these results, we can add them to the SM.

6. I'm confused how "mutual understanding" is correlated with INS over time (Fig. 4B). Was mutual understanding a scalar value for the entire task, and INS at each time point was correlated with that value across subjects? This choice needs to be justified more clearly. A better way to do this would be to have some continuous measure of understanding (e.g., ask people to transcribe what they heard after the task or continuously report understanding during the task) and correlate this throughout the task, since presumably mutual understanding varied over time.

A6: Immediately after each task, the participants were asked to assess the quality of their communication on a 5-point scale (for the LA pairs only). In this study it was assumed that the overall communication quality was established gradually and throughout the task. We did not ask the listeners to continuously report their understanding levels because this procedure would have interrupted the naturalistic communications. Alternatively, the participants could be asked to recall retrospectively what they had heard at a specific time point and how well they understood their partners, but the accuracy of such recalls after a complex multi-speaker task is probably of low reliability. Therefore, in our study the overall score was used to index the overall quality of communication between the listener and attended speaker. This score was then correlated with the INS increase at each time point to determine how early INS would predict the overall quality of communication. We have revised the METHODS and RESULTS to make it clearer (see lines 561-565).

That said, if it is possible in future studies to include on-line measurement of the understanding levels in the listener with minimal disruptions to the natural conversations, it would help to further clarify the relation between INS and the cocktail-party effect.

7. It's interesting that INS was best correlated with "mutual understanding" prior to a verbal response. The authors should cite and relate their findings to several studies (Liu et al, Scientific Reports, 2017; Stephens et al, 2010) that found strongest neural coupling at a lag in the time series of ~6-10 seconds (meaning that speakers' activity preceded listeners' activity by a few seconds, due

to anticipatory preparation of upcoming speech). How does this finding relate to this previous work (and can the authors replicate the lag correlation result with their data)? I'm also surprised the authors don't expect PFC to be involved in this process and even use it as a control region here. Please explain this discrepancy.

A7: First, findings from both Stephens et al. (2010) and Liu et al. (2017) suggest that neural synchronization is not related to simple speaking and listening behaviors, but rather to high-level cognitive functions such as prediction and comprehension^{10, 11}. To replicate these findings, we also shifted the time course of the listener relative to that of the speaker and vice versa from 1 s to 10 s (step = 1 s). Results showed that the highest increase of INS occurred when the brain activity of the listener preceded that of the speaker by 1-3 s (see lines 263-273, Figure 4). Thus, both the current and those of previous findings confirmed the involvement of high-level cognitive functions in interpersonal communications. The current findings additionally demonstrated this effect in a multi-speaker context.

Second, the reason that PFC was selected as a control region is as follows. Both TPJ and PFC have been found to be involved in communications, but they are associated with different cognitive functions. While PFC is more closely associated with attention, planning, and emotion^{12, 13, 14}, TPJ is more closely associated with the prediction of each other's subsequent actions (i.e., high-level mentalizing)¹⁵. In our study, we did not find any significant results on PFC (see lines 245-248, 259-262), which also suggested a dissociation of the roles of PFC and TPJ in social communications. Thus, to use PFC as a control would help to link the INS increase with mentalizing specifically rather than to general high-level cognitive functions.

References:

10. Liu Y, *et al.* Measuring speaker-listener neural coupling with functional near infrared spectroscopy. *Scientific reports* **7**, 43293 (2017).
11. Stephens GJ, Silbert LJ, Hasson U. Speaker-listener neural coupling underlies successful communication. *Proceedings of the National Academy of Sciences of the United States of America* **107**, 14425-14430 (2010).
12. Dixon ML, Thiruchselvam R, Todd R, Christoff K. Emotion and the Prefrontal Cortex: An Integrative Review. *Psychological bulletin*, (2017).
13. Hoshi E. Functional specialization within the dorsolateral prefrontal cortex: a review of anatomical and physiological studies of non-human primates. *Neuroscience research* **54**, 73-84 (2006).

14. Adolphs R. Social attention and the ventromedial prefrontal cortex. *Brain : a journal of neurology* **137**, 1572-1574 (2014).
15. Konvalinka I, Vuust P, Roepstorff A, Frith CD. Follow you, follow me: continuous mutual prediction and adaptation in joint tapping. *Quarterly journal of experimental psychology* **63**, 2220-2230 (2010).

8. Figure 1 is very poorly organized; the flow between graphics is not clear, and it should be completely re-vamped.

A8: Figure 1 has been removed.

9. Plot labels should be clearer (e.g., show “listener”/“speaker” on matrix axes)

A9: These labels have been modified to make them clearer (see Figure 2 and 3).

Reviewer #3 (Remarks to the Author):

In this paper the authors use FNIRS technology in order to measure neural activity simultaneously from 3 participants engaged in a “social context”, and attempt to quantify the inter-personal neural synchronization among them as a function of attention and communication quality. I find the design and experiment to be novel and timely, as it aims to address an issue that has long interested the cognitive neuroscience community regarding inter-personal communication.

This is also, however, a methodologically challenging feat, and requires many controls, which I am not sure are all properly addressed in the current manuscript.

My first concern is that when two people are engaged in a conversation, they share many common sensory inputs (primarily auditory) which would manifest in similar neural responses that also termed INS but would actually be trivial. The authors do not address this head-on, and it is a critical point before making claims about effects of attention and higher-order cognitive functions needed for communication.

A1: Thank you for the insightful comment. We excluded the shared sensory inputs as a possible explanation by showing that 1) The INS increase of the LA pairs was significantly stronger than that of the LU pairs even though the listeners shared the same sensory inputs with both the attended and unattended speakers (speaking rate, fluency, and naturalness, etc., were assessed and shown to be

comparable between the attended and unattended speakers, see A6 below); 2) The INS increased significantly prior to the occurrence of actual speaking behaviors, and was correlated significantly with communication quality; 3) The LA-LU differences in the INS increase was replicated in both the face-to-face and back-to-back conditions; 4) We added an analysis by shifting the time course of the listener forward relative to that of speaker by 1-10 s and vice versa and significant differences between the LA and LU pairs in the INS increase were found when the listener's brain activity preceded that of the speaker by 1-3 s (see A7 to Reviewer #2). Together, these findings suggest that the significant INS increase is not related to the shared sensory inputs such as articulatory movements or acoustic features of the speech signal, but rather likely to be associated with high-level information processing.

My second main concern is that monitoring neural activity in this type of naturalistic experiment makes it difficult to related specific neural response to behavior, as it is not controlled. While the authors clearly made an effort to code some aspects of the behavior diligently, clearly not all "talking" responses can be considered the same. It is critical to provide detailed information about the statistics of the behaviors monitored (see specific comments below).

A2: See our responses below to the specific comments.

My third main concern regards the comparison of face-to-face vs. back-to-back conditions. Clearly engaging in a face-to-face interaction, which involves multiple additional visual and motor cues is expected to increase INS as well as the efficacy behavioral communication. Yet in the presented data there seems to be no differences. This comparison should be more broadly explored.

A3: As this reviewer expected, our results actually demonstrated a difference, i.e., an additional increase of INS in the face-to-face as compared to the back-to-back conditions. The difference was found between pSTC of the listener and TPJ of the speaker (CH7-3), no matter whether there were multiple speakers or just a single speaker, and no matter whether the time points that corresponded to continuous speech of the listener was removed or not.

Following Reviewer #1's helpful suggestion, we additionally conducted FLDA to use INS increases to predict/classify different speakers. Different sets of CHs between the face-to-face and back-to-back conditions were found to have significant contributions to the high-level accuracy of prediction (see A4 to Reviewer #1). Together, these findings confirmed previously reported differences between the face-to-face and back-to-back communications (Jiang et al., 2012). In the revised manuscript, we have elaborated on these results.

Specific Comments:

Introduction (and abstract) should contain a more detailed explanation of what INS is - e.g. what aspects of neural activity are synchronized? How is it measured?

A4: Details about INS has been added to the ABSTRACT, INTRODUCTON, and the first paragraph of RESULTS. Also see A1 to Reviewer #2.

Methodological questions:

Where were the participants looking in the face-to-face condition? If the listener was looking at the attended speaker, this will obviously affect their neural synchronization (as well as speech-tracking responses in auditory cortex and other brain regions), and the comparison with the unattended speaker is no longer valid/comparable.

A5: In a naturalistic communication context, a listener would look at the face of the attended speaker. This was also true in the present study. However, the enhancement of INS in the LA pairs compared to LU pairs was consistently revealed in both the face-to-face and back-to-back conditions. This finding excluded the potential confounding effect of fixation direction on the conclusion.

Compared to the back-to-back condition, the face-to-face condition had an additional increase of INS between pSTC of the listener and TPJ of the speaker. This finding suggests a visual input effect on the increase of INS in the face-to-face condition (see A3 to this reviewer).

Were the participants given instructions about what to talk about? (e.g. what were the topics?)

Also, did the two “talkers” know which one was attended? Is it possible that if you know you are designated as an “attended” (and engage in a conversation) or “unattended” (speaking to yourself), this will change the way you talk (speed, content, enthusiasm etc.)

A6: Six topics were assigned to the 6 tasks (3 in the face-to-face and 3 in back-to-back conditions) in a counterbalanced way (see Methods). For each task, the same topic was used in the LA and LU pairs in order to control for the levels of familiarity and difficulty. In addition, although the attended and unattended speakers spoke about the same topic, the contents and specific sentences were different (the speakers had time to think about what they would say after getting the topic and relevant background material). We believe this design struck a balance between the need for naturalistic conversations and the need for comparable contents (also see A4 to Reviewer #2).

When the target speaker was assigned, the two speakers knew who was attended to before the onset of the task. When the target speaker was freely chosen by the listener, the speakers should be able to guess whether she/he was attended to based on the listener's fixation direction in the face-to-face condition or the listener's response in the back-to-back condition. In all of these tasks, the communication patterns of the two speakers such as speaking rate, fluency, and naturalness were assessed by the listener on a 5-point scale (1 represents the lowest level, and 5 represents the highest level) and shown to be comparable between the attended and unattended speakers (Mann-Whitney U test, $P > 0.05$, Šidák correction). In this revision, we additionally coded speaking rate and percentage of disfluency (i.e., repetition, pause, and interjection) by transcribing the speech of both speakers into texts based on the video of the multi-speaker conditions with assigned target. Statistical tests did not find any significant differences between the attended and unattended speakers [speaking rate: $t(40) = 0.934$, $P = 0.356$; percentage of disfluency: $t(40) = 1.598$; $P = 0.118$]. These results ensured the comparability of the communication patterns between the two speakers. These data have been added to the revised manuscript (see lines 130-134 in the main text, also see A10 to this reviewer).

Moreover, when the target speaker was freely chosen by the listeners, results also showed enhancement of INS at TPJ-TPJ, suggesting that even when the speaker knew that she/he was unattended to, she/he would speak as usual so that the listener could be attracted by her/him from time to time (for detailed comparisons between the two speakers in the chosen target tasks, please see A7 to this reviewer).

In the condition where the attended speaker was “freely chosen” by the listener - how did the coders determine who was attended in the back-to-back condition? Also, did the listener tend to select one speaker in this condition or to “jump” between them? And were the communication ratings and frequency of response different in this condition? Please provide some behavioral data on this.

A7: For the multi-speaker tasks when the target speaker was freely chosen by the listener, the attended speaker was determined by two independent coders based on the video recordings. Time points where the listener attended to each speaker were marked. The criteria of attention were as follows: 1) direction of the listener's face; 2) fixation on the target; 3) target of verbal and non-verbal responses. In the back-to-back condition, the determination was made based on the target of verbal and non-verbal responses only (see lines 444-457).

The inter-coder reliability (based on ICC) for switching (vs. no switching) was computed at the time-point level for each individual group. The ICC ranged from 0.833 to 1 (i.e., high consistency

between the two coders) in the face-to-face condition and from 0.566 to 0.633 in the back-to-back condition. The somewhat lower inter-coder consistency in the back-to-back condition was probably due to the unavailability of visual information such as the direction of the listener's face and fixation at the target. However, the ICC for overall number of switching computed at the group level was satisfactory to high in both conditions (0.996 in face-to-face; 0.970 in back-to-back). This information is now described in METHODS (see lines 444-457).

Listeners in 13 groups in the face-to-face condition and 11 groups in the back-to-back condition jumped between speakers, whereas listeners in the remaining groups did not. There were 3 types of evidence supporting the conclusion that the listener did not prefer a specific speaker. 1) An ANOVA was conducted on the overall lengths of periods when the listener attended to the two speakers in the face-to-face and back-to-back conditions. The independent variables were mode of communications (i.e., face-to-face vs. back-to-back) and target of speaker (Speaker 1 vs. Speaker 2). No significant effects were found (main effect for mode of communications: $F_{(1, 20)} = 0.223, P = 0.642$; main effect for the number of speakers: $F_{(1, 20)} = 0.102, P = 0.753$; interaction: $F_{(1, 20)} = 0.216, P = 0.647$). 2) Also, an ANOVA was conducted on scores of communication quality, and no significant effects were found for the main effect of the mode of communications ($F_{(1, 20)} = 0.061, P = 0.808$), the main effect for the number of speakers ($F_{(1, 20)} = 1.792, P = 0.195$), or the interaction effect ($F_{(1, 20)} = 3.152, P = 0.09$). 3) Finally, an ANOVA was conducted on the frequency of responses, and significant effects were found for the mode of communications ($F_{(1, 20)} = 26.842, P < 0.001$), type of responses ($F_{(1, 20)} = 17.041, P = 0.001$), but no significant effect was found for target of the speakers ($F_{(1, 20)} = 1.174, P = 0.291$), nor significant interactions (interaction between mode of communications and type of responses: $F_{(1, 20)} = 4.265, P = 0.052$, interaction between mode of communications and target of the speakers: $F_{(1, 20)} = 0.932, P = 0.346$; interaction between type of responses and target of the speakers: $F_{(1, 20)} = 0.173, P = 0.681$; 3-way interaction: $F_{(1, 20)} = 353, P = 0.559$). Together, these behavioral data did not suggest any preferences of the listener for a specific target speaker either in the face-to-face or in the back-to-back condition. These results have been added to the revised manuscript (see lines 161-165, 492-504 in the main text, and lines 30-47 in the supplementary materials, SM).

Also, please provide behavioral data on the face-to-face vs. back-to-back conditions

A8: The frequency of *non-verbal* responses differed significantly between the face-to-face and back-to-back conditions in both single-speaker and multi-speaker tasks, but the frequency of *verbal*

responses differed significantly between the face-to-face and back-to-back conditions only in the multi-speaker tasks with freely chosen target (see A5 to Reviewer #2 and A7 to this Reviewer).

In tasks with chosen target, no significant differences were found between the face-to-face and back-to-back conditions with regards to the length of attentional period to the target speakers and scores of communication quality (see A7 to this Reviewer).

Could both speakers hear the acoustic environment (i.e. what they and the other person were saying?)

A9: In both tasks with assigned target and freely chosen target, the two speakers could hear what she/he and the other speaker were saying. Thus, the acoustic environments were shared between the LA and LU pairs.

The Verbal behavioral measures coded indicate that the listener also spoke during the experiment (a point not mentioned when describing the experiment). Could you please add information regarding the communication and speech dynamics - what % of the time did each speaker speak, what was the dynamic of turn-taking/interjections, can you assess the speech fluency of each speaker (rate of speech, pauses etc); and conversely for the unattended speaker - did she have more pauses/less fluent speech than the attended+listener

A10: In the experiment, the listener was allowed to respond. This is consistent with a naturalistic cocktail-party context. This issue is discussed in METHODS (lines 388-409). The data showed that in the assigned target tasks, the listeners had continuous speech for about 10% and 11% of the 280s on average in the face-to-face and back-to-back conditions respectively. The speakers were speaking during the rest of the time. We also calculated the percentage of turn-takings and interjections. There were 11% and 9% verbal responses on average, and 5% and 1% nonverbal responses on average, in the face-to-face and back-to-back conditions respectively when the target was assigned. When the target was freely chosen, verbal responses were 8% for the face-to-face and 6% for the back-to-back conditions, whereas non-verbal responses were 6% for the face-to-face and 1% for the back-to-back conditions. For detailed statistical tests on frequency of response, please see A5 to Reviewer #2 and A7 to this Reviewer. The statistics did not show any significant differences between LA and LU in the frequency of responses by the listener.

Immediately after the end of the tasks, the listener was asked to assess several aspects of the communication patterns in Speaker 1 and Speaker 2 on a 5-point scale (1 represents the lowest level, and 5 represents the highest level). This assessment was based on the overall period of the task. The

assessment had two aspects, e.g., verbal and non-verbal communications. For verbal communications, 6 items were included: 1) Speed; 2) Loudness; 3) Fluency; 4) Naturalness of intonation; 5) Clarity; 6) Appropriateness in wording and syntax. The inter-item consistency reached a high level in both the face-to-face condition (Cronbach's alpha = 0.827 for Speaker 1 and 0.934 for Speaker 2) and the back-to-back condition (Cronbach's alpha = 0.893 for Speaker 1 and 0.910 for Speaker 2). The non-verbal aspect also had 6 items: 1) Naturalness; 2) Frequency of nodding; 3) Frequency of hand gestures; 4) Frequency of facial expressions; 5) Frequency of eye gaze; and 6) Frequency of smiling. The inter-item consistency was also high in the face-to-face (Cronbach's alpha = 0.8 for Speaker 1 and 0.851 for Speaker 2) and the back-to-back (Cronbach's alpha = 0.882 for Speaker 1 and 0.884 for Speaker 2) conditions. Thus, scores of the items were summed to have an overall score for verbal communication and an overall score for non-verbal communication. A comparison between Speaker 1 and Speaker 2 did not find any significant differences in either verbal or non-verbal communications, nor between the face-to-face and back-to-back conditions (Mann-Whitney U test, $P > 0.05$, Šidák correction).

In this revision, we additionally coded the speaking rate and the percentage of disfluency such as repetition, pause, and interjection by transcribing the speech of both speakers based on the video. Statistical tests did not find any significant differences between the attended and unattended speakers [speaking rate: $t(40) = 0.934$, $P = 0.356$; percent of disfluency: $t(40) = 1.598$; $P = 0.118$]. These results ensured the comparability of LA and LU pairs. These data have been added to the revised manuscript (see lines 2-29 in the SM).

Also, perhaps it is incorrect to refer to the participants as: listener, attended and unattended, but rather as one communicating pair and another individual talking to herself?

Was the analysis focused only on the parts of the session where she was silent?

A11: In a naturalistic cocktail-party context, it is natural for a listener to respond. Also, in the multi-speaker tasks with assigned target, the unattended speaker was also required to talk to the listener as if the listener was attending to her/him. Moreover, in the multi-speaker tasks with freely chosen target, the attention of the listener switched dynamically between the two speakers, and both speakers were trying to attract the attention of the listener, rather than to talk to herself/himself.

In addition, the continuous speech in the listeners occupied only about 10% of the total task period. Thus, for most periods of the tasks, the listener was a true listener. Also, we excluded the time points where the listener spoke continuously, and the results of the INS increase were consistent with those computed from all time points in all multi-speaker tasks (see A2 to Reviewer #1).

Therefore, to be consistent with previous studies of the cocktail-party effect^{8, 16, 17, 18, 19, 20}, listener and speaker are still the best options in the present study.

References:

8. Ding N, Simon JZ. Emergence of neural encoding of auditory objects while listening to competing speakers. *Proceedings of the National Academy of Sciences of the United States of America* **109**, 11854-11859 (2012).
16. Du Y, Kong L, Wang Q, Wu X, Li L. Auditory frequency-following response: a neurophysiological measure for studying the "cocktail-party problem". *Neuroscience and biobehavioral reviews* **35**, 2046-2057 (2011).
17. Zion Golumbic E, Cogan GB, Schroeder CE, Poeppel D. Visual input enhances selective speech envelope tracking in auditory cortex at a "cocktail party". *The Journal of neuroscience : the official journal of the Society for Neuroscience* **33**, 1417-1426 (2013).
18. Zion Golumbic EM, *et al.* Mechanisms underlying selective neuronal tracking of attended speech at a "cocktail party". *Neuron* **77**, 980-991 (2013).
19. Cherry A. Some experiments on the recognition of speech, with one and with two ears *The Journal of the acoustical society of America*, (1953).
20. Vander Ghinst M, *et al.* Left Superior Temporal Gyrus Is Coupled to Attended Speech in a Cocktail-Party Auditory Scene. *The Journal of neuroscience : the official journal of the Society for Neuroscience* **36**, 1596-1606 (2016).

Results

Is it possible to tease apart the aspects of the INS measure which pertain to encoding the acoustic input (which obviously both the listener and the speakers share; and therefore might be trivial), and aspect that are more selective and related to attention? The permutation test which mis-matched listeners and speakers from different sessions does not account to the inherently shared acoustic within-session...

A12: See A1 to this Reviewer.

Although the statistics were done on individual sensor-pairs, and few pairs passed the statistical threshold, the correlation matrices shown in Figure 3 suggest that the INS between L-attended is not really specific to one "pair" of sensors, but is actually broader and found in all condition. E.g., if you were to compare the mean INS in all pairs across conditions, it seems likely you would find patterns

not represented in the data as is. Perhaps a multi-voxel pattern analysis would be more appropriate, and/or aggregate statistics, rather than multiple individual t-test + corrections, which by definition reduce your ability to discover real effects.

A13: Thanks for the suggestions. We have recalculated the differences between the LA and LU pairs using two-sample *t*-tests either with FDR correction for all CHs or by basing the analyses on time points when the listener was silent. The results still showed significant differences between the LA and LU pairs at TPJ-TPJ consistently across various multi-speaker tasks. No other pairs of CHs survived these rigorous tests (see lines 142-158).

In addition, based on this reviewer and Reviewer #1's suggestions, we additionally conducted Fisher Linear Discriminate Analyses (FLDA) to address the concern that this reviewer raised. The averaged accuracy reached 82% for the LA pairs and 88% for the LU pairs (total accuracy = 85%) across all tasks. Most importantly, in each task, a different set of CHs showed significant contributions to the prediction, which confirmed the comments of this reviewer. However, only the CH of CH3-3 at TPJ-TPJ was shared by all multi-speaker tasks but not for the single speaker tasks (see lines 193-223 and Table 1). In sum, these findings confirmed that TPJ played a critical role the cocktail-party effect, and suggested that other brain areas may play supporting roles (also see A4 to Reviewer #1).

Minor

Abstract "while they communicated naturalistically" - should be rephrased

A14: This statement has been rephrased as "while they communicated".

Also, please refrain throughout the manuscript from referring to "the cocktail party effect" as a measure/manipulation, e.g. "the cocktail party effect was localized to TPJ " (line 163); "we expected a greater INS increase for the cocktail party effect" (p. 5)

A15: Thanks for the suggestions. We have revised the manuscript to avoid the use of this term as a measure or manipulation.

Reviewer #4 (Remarks to the Author):

The paper "selective enhancement of interpersonal neural synchronization underlies the cocktail

party effect” studies the synchronization between the neural signals in the brains of a listener and the attended and unattended speakers using fNIRS signals.

The authors show increased interpersonal neural synchrony (INS) in left TPJ between the listener and attended speaker which also correlates with the quality of listening. The authors suggest that INS may be the neural precursor of cocktail party effect.

The questions asked in the study are important, the experiments are well designed, and the analysis of the neural data is sound. However, I am quite uncertain about the implication of the findings and the claims that are made regarding “whether and how the INS underlie the extraction of the target speech”. Specifically:

- In general, it has been shown that the acoustic waveform that is produced by a speaker is highly correlated with the movement of the articulators (lips, jaw, tongue, ...), and the movement of the articulators is highly correlated with the neural activity in the speaker (e.g. Bouchard et. al. Nature 2013). The acoustic signal is also highly correlated with the neural activity in the brain of listener (Ding et. al., Zion-Gollumbic, et. al., ...), and many studies have shown that the neural signals in a listener track only the attended speaker (Ding et. al, Zion-Gollumbic et.al, O’Sullivan et. al. among many more).

Given all these, I am not sure why the findings of this study are novel or surprising, where what is shown is basically the tracking of the attended speaker in the listener’s brain. In other words, this could be explained simply by the auditory manifestation of speaker’s articulatory movements, and its encoding in the brain of a listener.

The only argument that the authors present in the paper against the hypothesis that the increased INS reflects merely auditory tracking is the delayed verbal response which comes later than the INS increase. However, the authors have no way of knowing when the subjects actually start tracking the attended speaker. It well may be that the listener attends and tracks a speaker for few seconds before giving a verbal or non-verbal response, therefore, it cannot be taken to indicate the lack of auditory tracking prior to the response.

A1: This reviewer is wondering what is new about this study if the acoustic waveform is synchronized between speakers and listeners. As our responses above (A1, 5,7, and 10 to Reviewer #3) showed, our study actually ruled out the shared sensory input hypotheses including the acoustic

waveform hypothesis and the articulatory and gesture hypothesis. More broadly, we believe that our results are significant for two main reasons. First, for the first time, we tested the hypothesis that INS has a role in the cocktail-party effect. The cocktail-party effect is a classic phenomenon whose neural mechanisms are not yet well understood. Previous research has focused on the acoustic features and their extraction, but our research aimed to clarify the role of interpersonal neural synchronization in solving the cocktail-party problem. We mentioned the extraction of acoustic features in our introduction, which might have misled Reviewer #4 to think that we wanted to focus on acoustic features. Instead we took the novel approach of focusing on the role of INS in the selective processing of relevant information while discounting the other information in a multi-speaker situation. Specifically, we proposed that neural synchronization between brains as a potential neural mechanism for the selective processing of information in a multi-speaker situation. Our hypothesis was based on previous studies that found associations between movement of articulators and brain activity in the speaker (brain-to-articulation)²¹, and between temporal features of speech sound and brain activity in the listeners (brain-to-speech)^{8, 9}. Although previously it has been speculated that there should be a relationship between the brain activity of the listener and that of the attended speaker in a multi-speaker situation. However, this hypothesis has never been tested in such a naturalistic context. We extended previous research to the associations between the listener and the speaker (brain-to-brain) and to test our INS hypothesis for the cocktail-party effect. No study thus far has examined the role of synchronization across brain areas in the cocktail-party effect.

Second, we have several pieces of evidence to support our conclusion that INS selectively processed high-level information rather than simple sensory information, which is also a novel finding. 1). We replicated the time-lag analysis that was used in previous studies^{10, 11}. Results showed that the highest enhancement of INS occurred when the brain activity of the listener preceded that of the speaker by 1-3 s (meaning that it was impossible for the listener's brain activity to track the acoustic speech signal of the speaker because the corresponding speech signal had not been produced by the speaker at that time point). 2). When there was no time-lag between the time courses of the speaker and listener, the highest level of INS increase occurred prior to the verbal response, and at this time point the INS increase was correlated significantly with communication quality, rather than the speaking patterns such as speaking rate, percent of disfluency, or frequency of responses. 3) Significant INS increases were replicated in both face-to-face and back-to-back conditions, which excluded the role of visual information in the selective process. 4). In the multi-speaker tasks with freely chosen target, the listener's attention dynamically switched between

the two speakers. A series of test of behavioral data between the two speakers did not find any significant differences (see A7 to Reviewer #3). Previous evidence has shown that semantic cues accelerate context updating and attention switching. Thus, it is more likely that the listener's attention was dynamically attracted by the contents, rather than the acoustic waveform of the speech signal from the other speaker. 5) The temporal analyses of the INS increase at TPJ-TPJ was repeated on that of PFC-PFC, but did not find any significant results (see A7 to Reviewer #2). Together, the evidence supported the conclusion that the increase of INS was not related to tracking the acoustic speech signal at any temporal scales or the visual articulatory or gestural movements. Instead, high-level information was involved. Moreover, the high-level information seemed to be associated with high-level mentalizing at TPJ rather than the general high-level cognitive functions at PFC.

References:

21. Park H, Kayser C, Thut G, Gross J. Lip movements entrain the observers' low-frequency brain oscillations to facilitate speech intelligibility. *eLife* **5**, e14521 (2016).

- The authors claim that their study supports the perspective that neural synchrony is a general mechanism underlying selective grouping of object features in a competing situation. The temporal coherence as a binding mechanism of acoustic features (which I assume is the one the authors are referring to) is very different from what the authors show in this study (< 0.4 Hz), it is very unclear how their finding provides supporting evidence for this mechanism.

A2: We are sorry for the misleading statements. The claim that "neural synchrony is a general mechanism underlying selective grouping of object features in a competing situation" has been changed to "Two lines of previous research have implicated the synchronization of neural activities within and between brains as a potential neural mechanism for the selective processing of information in a multi-speaker situation" in the INTRODUCTION and "providing supportive evidence for the perspective that neural synchronization is a potential neural mechanism for the selective processing of the attended information in a multi-speaker context" in the DISCUSSION. In addition, statements such as "the temporal coherence as a binding mechanism of acoustic features" has been removed as we did not focus on the acoustic features in this study.

- "The peak of the INS increase in the left TPJ likely marks the achievement of mutual understanding in the cocktail party effect.". I am not quite sure what they mean by mutual understanding when only one person talks the whole time (speaker vs. listener).

A3: Although in a naturalistic multi-speaker situation, the understanding of conversations should be mutual and we measured it as how well they understood each other, we agree with this reviewer that it would be more appropriate to label this as quality of communication because the communication of interest is mostly from the speaker to the listener.

- I also find statements like “INS increase in TPJ played a specific role in selectively extracting the target speech” not compelling since the authors haven’t actually shown a role for TPJ in extracting the target speech (only an increase in INS measure, which doesn’t necessarily mean a causal mechanism).

A4: We apologize for the confusion. We have avoided the causal statements and the inference about specific speech features.

- Increased TPJ during freely selecting target compared to fixed target is interesting, however, it is not clear what the authors mean by “listener had to hold all information”. Is it possible to actually show this? For example, do they perceive the unattended better in this situation compared to assigned target?

One point that casts doubt on this finding is the way the authors "guess" the attended speaker. One may argue that inaccuracies in specifying the actual attended speaker may partly explain this difference, since the metrics the authors used are not based on comprehension, but based on behavioral indicators such as the direction of the head or nodding patterns which inherently is not very accurate.

A5: The statement of “listener had to hold all information” was a speculative discussion about the findings that both the LA and LU pairs showed significant increase of INS at pSTC-pSTC. In this study there was no direct supportive evidence for this speculation. Furthermore, after correcting for multiple comparisons involving all CHs in the new analysis, the relevant result was no longer significant. We have removed this speculation.

Minor: Arnott et. al. 2004 reference is missing.

A6: This citation has been removed.

Full list of the references

1. Cui X, Bray S, Bryant DM, Glover GH, Reiss AL. A quantitative comparison of NIRS and fMRI across multiple cognitive tasks. *Neuroimage* **54**, 2808-2821 (2011).
2. Scholkmann F, *et al.* A review on continuous wave functional near-infrared spectroscopy and imaging instrumentation and methodology. *NeuroImage* **85, Part 1**, 6-27 (2014).
3. Gagnon L, Perdue K, Greve DN, Goldenholz D, Kaskhedikar G, Boas DA. Improved recovery of the hemodynamic response in diffuse optical imaging using short optode separations and state-space modeling. *Neuroimage* **56**, 1362-1371 (2011).
4. Gagnon L, Cooper RJ, Yucel MA, Perdue KL, Greve DN, Boas DA. Short separation channel location impacts the performance of short channel regression in NIRS. *Neuroimage* **59**, 2518-2528 (2012).
5. Gagnon L, Yucel MA, Boas DA, Cooper RJ. Further improvement in reducing superficial contamination in NIRS using double short separation measurements. *Neuroimage* **85 Pt 1**, 127-135 (2014).
6. Lu CM, Zhang YJ, Biswal BB, Zang YF, Peng DL, Zhu CZ. Use of fNIRS to assess resting state functional connectivity. *Journal of neuroscience methods* **186**, 242-249 (2010).
7. Zhang YJ, Lu CM, Biswal BB, Zang YF, Peng DL, Zhu CZ. Detecting resting-state functional connectivity in the language system using functional near-infrared spectroscopy. *Journal of biomedical optics* **15**, 047003 (2010).
8. Ding N, Simon JZ. Emergence of neural encoding of auditory objects while listening to competing speakers. *Proceedings of the National Academy of Sciences of the United States of America* **109**, 11854-11859 (2012).
9. Mesgarani N, Chang EF. Selective cortical representation of attended speaker in multi-talker speech perception. *Nature* **485**, 233-236 (2012).

10. Liu Y, *et al.* Measuring speaker-listener neural coupling with functional near infrared spectroscopy. *Scientific reports* **7**, 43293 (2017).
11. Stephens GJ, Silbert LJ, Hasson U. Speaker-listener neural coupling underlies successful communication. *Proceedings of the National Academy of Sciences of the United States of America* **107**, 14425-14430 (2010).
12. Dixon ML, Thiruchselvam R, Todd R, Christoff K. Emotion and the Prefrontal Cortex: An Integrative Review. *Psychological bulletin*, (2017).
13. Hoshi E. Functional specialization within the dorsolateral prefrontal cortex: a review of anatomical and physiological studies of non-human primates. *Neuroscience research* **54**, 73-84 (2006).
14. Adolphs R. Social attention and the ventromedial prefrontal cortex. *Brain : a journal of neurology* **137**, 1572-1574 (2014).
15. Konvalinka I, Vuust P, Roepstorff A, Frith CD. Follow you, follow me: continuous mutual prediction and adaptation in joint tapping. *Quarterly journal of experimental psychology* **63**, 2220-2230 (2010).
16. Du Y, Kong L, Wang Q, Wu X, Li L. Auditory frequency-following response: a neurophysiological measure for studying the "cocktail-party problem". *Neuroscience and biobehavioral reviews* **35**, 2046-2057 (2011).
17. Zion Golumbic E, Cogan GB, Schroeder CE, Poeppel D. Visual input enhances selective speech envelope tracking in auditory cortex at a "cocktail party". *The Journal of neuroscience : the official journal of the Society for Neuroscience* **33**, 1417-1426 (2013).
18. Zion Golumbic EM, *et al.* Mechanisms underlying selective neuronal tracking of attended speech at a "cocktail party". *Neuron* **77**, 980-991 (2013).

19. Cherry A. Some experiments on the recognition of speech, with one and with two ears
The Journal of the acoustical society of America, (1953).
20. Vander Ghinst M, *et al.* Left Superior Temporal Gyrus Is Coupled to Attended Speech in a Cocktail-Party Auditory Scene. *The Journal of neuroscience : the official journal of the Society for Neuroscience* **36**, 1596-1606 (2016).
21. Park H, Kayser C, Thut G, Gross J. Lip movements entrain the observers' low-frequency brain oscillations to facilitate speech intelligibility. *eLife* **5**, e14521 (2016).

Reviewers' comments:

Reviewer #1 (Remarks to the Author):

The authors have been very responsive to both editing suggestions and substantive scientific critiques.

Reviewer #2 (Remarks to the Author):

The authors have responded to most of my main criticisms, but I would still like them to elaborate (maybe in the Discussion) on the implications of the distinct roles of the PFC and TPJ in the cocktail party effect. This is an effect of selective attention, but the authors argue that they expect the TPJ, which is more involved in "high-level mentalizing", to drive this process more than the PFC, "which is more closely associated with attention, planning and emotion". Given that previous studies (mentioned in my previous review) have shown that neural coupling in both TPJ and PFC relate to successful communication, I think the authors need to more clearly justify their use of PFC as a control region rather than an important component of this attentional tracking.

Reviewer #3 (Remarks to the Author):

I found the authors responses to several of my comments to be satisfactory. However, I still have some remaining questions regarding the multi-speaker task with freely chosen target. While I appreciate the difficulty in determining "where" attention was focused when the listener could freely shift between speakers, and the authors have attempted to deal with this rigorously, I still feel that the "average" results might not fully represent/characterize the variety of behaviors that this condition invites.

First, based on the authors response, it seems that approximately half of the groups attention "jumped" between speakers, and in half one speaker was chosen each trial. This dynamic can substantially change the type of listening, and I am not sure that pooling both types of behaviors into one analysis is the best way to go.

Second, I was glad to see that there was high agreement between the two coders about when "jumping" occurred in the face-to-face condition. However, the low correlation in the back-to-back condition leaves much room for doubt regarding where attention actually was at a given time point (which is critical for segmenting the data according to when each speaker was "attended"). It is also not clear to me what was done for time-points where there wasn't agreement between the coders as to which speaker was attended. The correlation of "overall number of switches" is less relevant for this analysis, as it is global but does not help with precise segmentation according to attention. Beyond my personal curiosity in seeing the actual rates of switching in this task, I think the "cleanest" way to address this caveat would be to use only the parts of the data where there was agreement between the coders, and perhaps excluded trials where there was a lot of switching.

Third, in their response the authors report an ANOVA done on the results of the coding in order to determine whether there is a preference to a particular speaker. In this analysis "target speaker" is used as an independent variable (Speaker 1 vs. Speaker 2). However, this seems quite irrelevant in the current design, since these are arbitrary categories and not "the same" across individuals. (e.g. each participant could indeed have one "favorite", but as long as this is different across participant, the statistics will not reflect this).

The same is true for all other analyses where Speaker 1 and Speaker 2 are directly compared (e.g. scoring of verbal and non-verbal communications), since these are not “really” different conditions.

Reviewer #4 (Remarks to the Author):

The revision has a better interpretation of results and more focused claims. I, however, am a bit confused by the insistence of the authors that the previously shown auditory tracking has nothing to do with the INS measure. Since the “only” signal that reaches the listener, at least in the back to back situation, is the sound of the attended speaker, any other underlying internal representation that the authors propose “has” to be extracted from the acoustic and therefore will be correlated with it. The studies that the authors cite all show strong tracking in the brain of the listener to the attended speaker. The answers the authors provide aren’t compelling.

- The observation that the listener precedes the speaker can be explained by predictive and contextual effects in speech. Speech is a very redundant signal in time, especially the extremely slow time-scales used here.
- Of course the auditory tracking only occurs for the attended speaker and not the unattended, so the difference between INS attended unattended is completely predictable from acoustics.
- Attended auditory tracking has also been shown to correlate with the degree of subject’s attention in studies they cited, so behavior dependence of INS is not an evidence against auditory tracking.
- The significance of the effect in both face to face and back to back actually suggests that the auditory tracking is the dominant factor, rather than eliminating it.
- The claim that the effect is high level semantic is not compelling, because the authors didn’t test this hypothesis and it’s mere speculation.
- And finally, the fact that they find a stronger effect in temporal vs. frontal also supports the evidence of acoustic tracking, as this previously cited studies have reported auditory tracking in Wernicke’s area and not frontal cortex.

While the authors are correct that brain to brain correlation has not been studied before for cocktail party (but in my view, predictable from previous studies), I don’t see compelling evidence to rule out the role of acoustic tracking in INS. Again, what else can it be after all, when acoustic is the only signal that reaches the listener? Any other “internal object” that could explain INS measures has to be extracted from acoustic since there is no other input, and hence it will be correlated with the acoustic and not differentiable using your method. This needs to be addressed more properly in the discussion.

Response to reviewers' comments

Reviewers' comments:

Reviewer #1 (Remarks to the Author):

The authors have been very responsive to both editing suggestions and substantive scientific critiques.

A1: Thank you very much.

Reviewer #2 (Remarks to the Author):

The authors have responded to most of my main criticisms, but I would still like them to elaborate (maybe in the Discussion) on the implications of the distinct roles of the PFC and TPJ in the cocktail party effect. This is an effect of selective attention, but the authors argue that they expect the TPJ, which is more involved in "high-level mentalizing", to drive this process more than the PFC, "which is more closely associated with attention, planning and emotion". Given that previous studies (mentioned in my previous review) have shown that neural coupling in both TPJ and PFC relate to successful communication, I think the authors need to more clearly justify their use of PFC as a control region rather than an important component of this attentional tracking.

A1: We have elaborated this issue in the Discussion. Specifically, the LA pairs showed a significant increase of INS at pSTC-PFC and pSTC-TPJ, but not at TPJ-TPJ, in the single-speaker tasks. Interestingly, the INS increase at pSTC-PFC and pSTC-TPJ also showed a relatively high level INS in both the LA and LU pairs in the multi-speaker context, although the level of INS at the former CH combination did not reach significance after FDR correction ($P < 0.05$). This result was consistent with previous reports that both PFC and TPJ were involved in successful communications in a clear auditory context⁴, and that successful communications not only involved the INS that occurred between the same brain areas⁴, but also that occurred between different brain areas⁵ (between the primary auditory cortex of the listener and high-level regions such as PFC and TPJ of the speaker in the present study). That is, the listener might access the linguistic information that was represented in the PFC and TPJ of the speaker by processing the sound inputs at the primary auditory areas.

In the multi-speaker tasks, the LA pairs additionally showed a significant INS increase at TPJ-TPJ, suggesting that high-level mentalizing⁶ is more important than merely access linguistic information in a noisy context.

Reviewer #3 (Remarks to the Author):

I found the authors responses to several of my comments to be satisfactory.

A1: Thank you very much.

However, I still have some remaining questions regarding the multi-speaker task with freely chosen target. While I appreciate the difficulty in determining “where” attention was focused when the listener could freely shift between speakers, and the authors have attempted to deal with this rigorously, I still feel that the “average” results might not fully represent/characterize the variety of behaviors that this condition invites.

A2: See below.

First, based on the authors response, it seems that approximately half of the groups attention “jumped” between speakers, and in half one speaker was chosen each trial. This dynamic can substantially change the type of listening, and I am not sure that pooling both types of behaviors into one analysis is the best way to go.

A3: We examined the INS increase at TPJ-TPJ separately for groups with or without switching between two speakers. The LA pairs had significantly higher INS increase than did the LU pairs at TPJ-TPJ for both types of groups: for the switching groups [face-to-face: $N = 13$, $t(12) = 4.036$, $P = 0.002$; back-to-back: $N = 11$, $t(10) = 3.192$, $P = 0.01$] and for the no switching groups [face-to-face: $N = 8$, $t(7) = 2.725$, $P = 0.03$; back-to-back $N = 10$, $t(9) = 2.341$, $P = 0.044$]. These new results have been added to the manuscript.

Besides, we also examined the INS that corresponded to the switching of the target in order to determine whether target switching modulated the increase of INS. Two types of switches were identified, i.e., switching from the left speaker to the right speaker and vice versa. This analysis also focused on TPJ (CH3) only. The increase of INS before, during, or after the switch did not differ from the INS increase at the other time points (Fig. 1 for this response letter). Moreover, we did not find any patterns similar as those associated with communicating response. This result was not added to the manuscript as it was irrelevant to the main findings.

Thus, it seems that the switching between targets weakened but did not modulate the level of INS.

Fig. 1 Results of the INS increase associated with switching of the target. (A) and (B) demonstrate results of switching from the left speaker to the right speaker, whereas (C) and (D) demonstrates the results of switching from the right speaker to the left speaker. Note that none of the INS increase reached significance compared to those when there was no switching.

Second, I was glad to see that there was high agreement between the two coders about when “jumping” occurred in the face-to-face condition. However, the low correlation in the back-to-back condition leaves much room for doubt regarding where attention actually was at a given time point (which is critical for segmenting the data according to when each speaker was “attended”). It is also not clear to me what was done for time-points where there wasn’t agreement between the coders as to which speaker was attended. The correlation of “overall number of switches” is less relevant for this analysis, as it is global but does not help with precise segmentation according to attention.

A4: First, the pattern of the INS increases in the back-to-back condition replicated that in the face-to-face condition, as well as that when the targets were *a priori* assigned. Thus, although there were probably more errors in coding the attended target in the back-to-back condition than in the face-to-face condition when the listener freely chose the target, it seems that these errors only weakened but did not change the pattern of the results. But it is certainly worth a further check whether there were differences between the back-to-back and face-to-face conditions when more reliable coding indicators are available in the future.

Second, for time-points where there was not an agreement between coders as to which speaker was attended, the two coders would discuss together until an agreement was obtained. The analyses were conducted based on data with final agreement between the two coders. We have added this information to the manuscript.

Beyond my personal curiosity in seeing the actual rates of switching in this task, I think the “cleanest” way to address this caveat would be to use only the parts of the data where there was agreement between the coders, and perhaps excluded trials where there was a lot of switching.

A5: Yes, the analyses were conducted all based on data with final agreement between the two coders.

We have separately analyzed the data with attention shift and those without attention shift. See A1 in this reviewer.

Third, in their response the authors report an ANOVA done on the results of the coding in order to determine whether there is a preference to a particular speaker. In this analysis “target speaker” is used as an independent variable (Speaker 1 vs. Speaker 2). However, this seems quite irrelevant in the current design, since these are arbitrary categories and not “the same” across individuals. (e.g. each participant could indeed have one “favorite”, but as long as this is different across participant, the statistics will not reflect this).

The same is true for all other analyses where Speaker 1 and Speaker 2 are directly compared (e.g. scoring of verbal and non-verbal communications), since these are not “really” different conditions.

A4: We are sorry for this misleading expression. We actually wanted to use spatial location (i.e., left or right relative to the listener) as the independent variable. This is because although the two speakers’ roles were randomly assigned across different groups, the spatial locations relative to the listener were fixed. However, our results did not show any preference for the spatial locations.

Reviewer #4 (Remarks to the Author):

The revision has a better interpretation of results and more focused claims. I, however, am a bit confused by the insistence of the authors that the previously shown auditory tracking has nothing to do with the INS measure. Since the “only” signal that reaches the listener, at least in the back to back situation, is the sound of the attended speaker, any other underlying internal representation that the authors propose “has” to be extracted from the acoustic and therefore will be correlated with it. The studies that the authors cite all show strong tracking in the brain of the listener to the attended speaker. The answers the authors provide aren’t compelling.

A1: We thank this reviewer for her/his insightful question. Sound input is the first and necessary step for speech comprehension. However, one recent study of EEG-based

hyperscanning specifically demonstrated that the existence of brain-to-brain synchronization is not merely an epiphenomenon of auditory processing during oral narratives between the listener and speaker ⁷. That is, brain-to-brain synchronization could not be explained by brain-audio envelope synchronizations ⁷.

In the present study, to directly exclude the possibility that brain-to-speech synchronization contributed to brain-to-brain synchronization, brain-to-speech synchronization and its relationship with brain-to-brain synchronization were also examined. Previously the hemodynamic signal has never been analyzed in brain-to-speech synchronization because of its sluggish nature. However, recent BOLD-effect evidence indicates that higher frequency fluctuations in BOLD signals (up to 0.8 Hz) also has neural relevance and makes functional contribution ^{8, 9}. Most importantly, evidence shows reconstruction of real-life sounds from fMRI response patterns (< ~2Hz) in the human auditory cortex ¹⁰. It is known that in BOLD signals, while low-frequency fluctuations are associated with the general excitability ¹¹ and spatially overlapped neural networks ¹², high-frequency fluctuations are associated with focal functions and can be a more direct and precise index of the cognitive processes ⁸. Most importantly, the auditory cortex can be reliably detected with the high frequency BOLD signal around 0.5-0.8 Hz ⁹. Thus, it is possible to evaluate brain-to-speech synchronization using the fNIRS signal. This analysis was conducted on the multi-speaker condition with assigned target.

To calculate the synchronization between brain activity and the speech envelope, the following steps were conducted. The experimental videos were converted into audios first. Because one of the DVs was connected to the ETG-4000 equipment, it was able to be synchronized with the fNIRS data acquisition. By examining the video content, the exact starting and ending time of the task was also marked on the video of the other DV. Thus, the converted audios were exactly synchronized with the fNIRS data. Next, a blind source separation (BSS) procedure ¹³ was conducted using an independent component analysis program of the EEGLAB software (version 14) ^{14, 15}. Briefly, in a multi-speaker situation, if each of the M speeches were recorded by N microphones, the recordings will be a matrix composed of a set of N vectors, each of which is a (weighted) linear superposition of the M speeches. BSS enables extraction of the target speech from observed mixed speech without the need for source positioning, spectral construction, or a mixing system. One of the key methods for performing BSS is ICA, where we can take advantage of (an assumed) linear independence between the sources. In this study, two DV-linked microphones were placed in two separate positions to record the speeches of each speaker. Based on this, the ICA program produced 2 ICs for each participant group. The ICs were labeled and saved as attended and unattended speeches respectively, according to the cross-checking of the contents between the ICs and the video. For each of the separated speeches, the amplitude envelope was computed using the envelope function of MATLAB and resampled to 10Hz (the same as the fNIRS data). Then, WTC was applied to compute the coherence values between the listener's fNIRS signal and the attended and unattended speeches, and between the two speakers' fNIRS signal and their speeches, in the multi-speaker task with assigned

target speaker in both face-to-face and back-to-back mode. Meanwhile, this computation was also conducted between the fNIRS signal at the resting-state and the speeches in the tasks, which were used as control data as no brain-to-speech synchronization was expected. Finally, paired two-sample t-test was conducted to examine whether there was significant brain-to-speech synchronization in the multi-speaker tasks compared to the control data, and whether the brain-to-speech synchronization, if there was any, differed significantly between the LA and LU pairs along the full frequency range (0.01-0.8 Hz). Data above 0.8 Hz were not considered according to previous BOLD evidence^{8, 9}. Because there were no significant results after FDR correction across all measurement CH-speech pairs over the full frequency ranges (11 pairs in total, 0.01-0.8Hz) at $P < 0.05$ level, uncorrected results at a significance level of $P < 0.0005$ were reported. To further exclude the potential influence of the CHs' brain-to-speech synchronization brain-to-brain synchronization, the INS increases for CH combinations related to brain-to-speech synchronization were averaged and used as the covariates when comparing the INS increase between the LA and LU pairs.

As there were no significant results after FDR correction across all measurement CH-speech pairs over the full frequency ranges (11 pairs in total, 0.01-0.8Hz) at $P < 0.05$ level, uncorrected results at a significance level of $P < 0.0005$ were reported. As expected, only the LA pairs had significant brain-to-speech synchronization at CHs that covered the left auditory cortices in both the face-to-face (f2f) and back-to-back (b2b) conditions. Specifically, in the f2f condition, the highest level of brain-to-speech synchronization in the LA pairs appeared at CH5 and 8-10 that covered the left auditory temporal cortex at a frequency range of around 0.8 Hz [CH5: $t(14) = 4.425$, $P = 0.00049$; CH8: $t(14)=4.45$, $P = 0.00047$; CH9: $t(14) = 4.862$, $P = 0.00021$; CH10: $t(14) = 4.917$, $P = 0.00019$]. No CHs survived the thresholding at any frequency ranges in the LU pairs ($P > 0.0005$). In the b2b condition, again, CH5 at the auditory temporal cortex had significant brain-to-speech synchronization at the same frequency range as that in the f2f condition (i.e., around 0.8 Hz) [$t(14) = 4.618$, $P = 0.0004$]. The b2b condition had an additional result at CH1 that covered the left inferior frontal cortex at a lower frequency range of around 0.05 Hz [$t(14) = 4.569$, $P = 0.00044$]. No significant results were found in the LU pairs. Also, no significant difference was found between the LA and LU pairs ($P > 0.0005$). These results confirmed previous reports based on EEG/MEG that the activity in the auditory cortex selectively tracks the attended speech while ignores the unattended one. However, such an effect did not overlap with the brain-to-brain synchronization in both anatomical locations and frequency range of the brain activity.

In addition, significant brain-to-speech synchronization was also found between the brain activity of the speaker and her/his speech in both the f2f and b2b conditions. That was, in the f2f condition, the consistent result between the two speakers appeared at CH4 that covered the left inferior frontal cortex at the same frequency range as the LA pairs (i.e., around 0.8 Hz) [left speaker: $t(14) = 4.667$, $P = 0.0003$; right speaker: $t(14) = 4.563$, $P = 0.00048$]. In the b2b condition, the consistent result between the two speakers also appeared at IFC at a frequency range of around 0.8 Hz [left speaker, CH1: $t(14) = 4.644$, $P = 0.00038$;

right speaker, CH4: $t(14) = 5.429, P < 0.0001$]. The b2b condition had an additional result at the auditory temporal cortex at a frequency range of around 0.5 Hz [left speaker, CH8: $t(14) = 5.08, P = 0.00017$; right speaker: CH10: $t(14) = 5.103, P = 0.00016$]. Again, TPJ (CH3) did not show significant brain-to-speech synchronization in the speakers.

To further exclude the potential influence of these CHs' brain-to-speech synchronization brain-to-brain synchronization at TPJ, the INS increases for CH combinations between CH4 and all other 11 CHs were averaged. Similar procedures were conducted on CH5, 8-10. These averaged INS increases were used as the covariates when comparing the INS increase of TPJ (CH3) between the LA and LU pairs using a repeated measure ANCOVA. Results still showed significantly higher INS increase in the LA pairs than in the LU pairs ($F_{(1,15)} = 7.056, P = 0.018$) in the f2f condition. Also, after covarying the INS increase related to CH1,4,5,8, and 10, the INS increase was still significantly higher in the LA pairs than in the LU pairs in the back-to-back condition ($F_{(1,15)} = 5.588, P = 0.032$).

In sum, these results excluded the possibility that the selectively enhanced INS at TPJ was contributed by brain-to-speech synchronization either in the listener or the speaker. These analyses, results, and discussions have been added to the manuscript.

- The observation that the listener precedes the speaker can be explained by predictive and contextual effects in speech. Speech is a very redundant signal in time, especially the extremely slow time-scales used here.

A2: There is no evidence demonstrating that speech redundancy leads to INS when the listener's brain activity precedes that of the speaker. On the contrary, there is evidence showing that linguistic structures expand in a unit of several seconds, whereas sound-level acoustic features expand in a unit of million seconds¹⁶. Thus, such time-lag results are more likely to reflect linguistic rather than acoustic processing.

- Of course the auditory tracking only occurs for the attended speaker and not the unattended, so the difference between INS attended unattended is completely predictable from acoustics.

A3: See A1.

- Attended auditory tracking has also been shown to correlate with the degree of subject's attention in studies they cited, so behavior dependence of INS is not an evidence against auditory tracking.

A4: It is not our purpose to provide evidence in support of or against auditory tracking. See A1.

- The significance of the effect in both face to face and back to back actually suggests that the auditory tracking is the dominant factor, rather than eliminating it.

A5: See A1.

- The claim that the effect is high level semantic is not compelling, because the authors didn't test this hypothesis and it's mere speculation.

A6: We have acknowledged that this conclusion was achieved by comparing the present results with those of previous studies.

- And finally, the fact that they find a stronger effect in temporal vs. frontal also supports the evidence of acoustic tracking, as this previously cited studies have reported auditory tracking in Wernicke's area and not frontal cortex.

A7: We found selectively enhanced INS at TPJ rather than Wernicke's area. On the contrary, we identified a pattern of relatively high-level increase of INS at pSTC-PFC and pSTC-TPJ in the LA pairs during the single-speaker tasks and in both the LA and LU pairs during the multi-speaker tasks, which might reflect sound processing. The brain-to-speech synchronization analyses also confirmed this conclusion.

While the authors are correct that brain to brain correlation has not been studied before for cocktail party (but in my view, predictable from previous studies), I don't see compelling evidence to rule out the role of acoustic tracking in INS. Again, what else can it be after all, when acoustic is the only signal that reaches the listener? Any other "internal object" that could explain INS measures has to be extracted from acoustic since there is no other input, and hence it will be correlated with the acoustic and not differentiable using your method. This needs to be addressed more properly in the discussion.

A8: See A1. In addition, we have added discussions about this issue to the manuscript.

References

1. Park H, Kayser C, Thut G, Gross J. Lip movements entrain the observers' low-frequency brain oscillations to facilitate speech intelligibility. *eLife* **5**, e14521 (2016).
2. Ding N, Simon JZ. Emergence of neural encoding of auditory objects while listening to competing speakers. *Proceedings of the National Academy of Sciences of the United States of America* **109**, 11854-11859 (2012).
3. Mesgarani N, Chang EF. Selective cortical representation of attended speaker in multi-talker speech perception. *Nature* **485**, 233-236 (2012).

4. Stephens GJ, Silbert LJ, Hasson U. Speaker-listener neural coupling underlies successful communication. *Proceedings of the National Academy of Sciences of the United States of America* **107**, 14425-14430 (2010).
5. Liu Y, *et al.* Measuring speaker-listener neural coupling with functional near infrared spectroscopy. *Scientific reports* **7**, 43293 (2017).
6. Konvalinka I, Vuust P, Roepstorff A, Frith CD. Follow you, follow me: continuous mutual prediction and adaptation in joint tapping. *Quarterly journal of experimental psychology* **63**, 2220-2230 (2010).
7. Perez A, Carreiras M, Dunabeitia JA. Brain-to-brain entrainment: EEG interbrain synchronization while speaking and listening. *Scientific reports* **7**, 4190 (2017).
8. Chen JE, Glover GH. BOLD fractional contribution to resting-state functional connectivity above 0.1 Hz. *Neuroimage* **107**, 207-218 (2015).
9. Gohel SR, Biswal BB. Functional integration between brain regions at rest occurs in multiple-frequency bands. *Brain connectivity* **5**, 23-34 (2015).
10. Santoro R, *et al.* Reconstructing the spectrotemporal modulations of real-life sounds from fMRI response patterns. *Proceedings of the National Academy of Sciences of the United States of America* **114**, 4799-4804 (2017).
11. Raichle ME. The restless brain. *Brain connectivity* **1**, 3-12 (2011).
12. Smith SM, *et al.* Temporally-independent functional modes of spontaneous brain activity. *Proceedings of the National Academy of Sciences of the United States of America* **109**, 3131-3136 (2012).
13. Comon P, Jutten C. *Handbook of Blind Source Separation: Independent component analysis and applications*. Academic press. (2010).
14. Delorme A, Makeig S. EEGLAB: an open source toolbox for analysis of single-trial EEG dynamics including independent component analysis. *Journal of neuroscience methods* **134**, 9-21 (2004).
15. Makeig S, Debener S, Onton J, Delorme A. Mining event-related brain dynamics. *Trends in cognitive sciences* **8**, 204-210 (2004).
16. Hasson U, Chen J, Honey CJ. Hierarchical process memory: memory as an integral component of information processing. *Trends in cognitive sciences* **19**, 304-313 (2015).

Reviewers' comments:

Reviewer #2 (Remarks to the Author):

The authors have made great efforts to address the reviewers' remaining concerns, especially regarding explanations based on brain-to-speech synchronization.

A figure comparing significant brain-to-brain and brain-to-speech channels would be helpful.

On p. 21, line 313, it is not correct to say that "the hemodynamic signal has never been analyzed in brain-to-speech synchronization because of its sluggish nature". Maybe I'm not sure exactly what you mean, but several papers, including Silbert, Honey, Simony, Poeppel, and Hasson (2014), have done this, so I would eliminate that statement.

Reviewer #3 (Remarks to the Author):

I have read the authors revised version of the manuscript and their response to my previous concerns. However, I still find that the condition where participants could (and did) spontaneously switch attention between speakers is not sufficiently robust.

The authors have added a figure to their response letter showing "no significant difference in INS" around the time of a switch. I find it very difficult to understand this data:

- a) First, how is the graph so "smooth" around the time of shifting, if the INS of the listener is now allegedly calculated relative to a different speaker?
- b) Although no systematic difference in INS was found before vs. after the shift, the dynamics shown in the sample graphs suggest that the INS is far from a stable response, and rather fluctuates regardless of attention switches as well.

My suggestion would be to exclude the spontaneous condition from the manuscript, since the different dynamics of switching across individuals make it inherently difficult to interpret and generalize from.

Response to reviewers' comments

Reviewers' comments:

Reviewer #2 (Remarks to the Author):

The authors have made great efforts to address the reviewers' remaining concerns, especially regarding explanations based on brain-to-speech synchronization.

A figure comparing significant brain-to-brain and brain-to-speech channels would be helpful.

Answer1(A1): Fig. 7 has been added to show CH combinations that had significant brain-to-brain synchronization and CH-speech pairs that had significant brain-to-speech synchronization.

On p. 21, line 313, it is not correct to say that "the hemodynamic signal has never been analyzed in brain-to-speech synchronization because of its sluggish nature". Maybe I'm not sure exactly what you mean, but several papers, including Silbert, Honey, Simony, Poeppel, and Hasson (2014), have done this, so I would eliminate that statement.

A2: We actually meant that no studies have examined the effect of acoustic tracking using the hemodynamic signal. But we found that a recent study by Santoro et al. (2017, PNAS) showed reconstruction of real-life sounds from fMRI response patterns (< ~2 Hz) in the human auditory cortex. Thus, this sentence has been deleted.

Reviewer #3 (Remarks to the Author):

I have read the authors revised version of the manuscript and their response to my previous concerns. However, I still find that the condition where participants could (and did) spontaneously switch attention between speakers is not sufficiently robust.

The authors have added a figure to their response letter showing "no significant difference in INS" around the time of a switch. I find it very difficult to understand this data:

- a) First, how is the graph so "smooth" around the time of shifting, if the INS of the listener is now allegedly calculated relative to a different speaker?
- b) Although no systematic difference in INS was found before vs. after the shift, the

dynamics shown in the sample graphs suggest that the INS is far from a stable response, and rather fluctuates regardless of attention switches as well.

My suggestion would be to exclude the spontaneous condition from the manuscript, since the different dynamics of switching across individuals make it inherently difficult to interpret and generalize from.

A1: According to this reviewer's suggestion, we have deleted the multi-speaker tasks when the targets were freely chosen. We have explained this in the Results (line 100-102) as follows: "Each group performed 3 communication tasks within each of the 2 experimental conditions. One communication task was excluded (see Methods and Supplementary Methods for the exclusion of one task)". We also explained this in the Methods (line 466-469) as follows: "Due to the low inter-judge reliability in determining the target speaker in a multi-speaker task with freely chosen target (see Supplementary Methods), this task was excluded from this study, leaving 2 experimental conditions by 2 communication tasks." Details about this task are provided in Supplementary Methods under the section of "Exclusion of the multi-speaker tasks with freely chosen target speaker". Accordingly, Figs. 2 and 3 were modified, and the ANOVA and FLDA on the INS increase were re-conducted (but the results did not change) in accordance to this change. We also carefully checked the whole manuscript to ensure consistency.

We thank both the editor and the reviewers for their encouraging comments and helpful suggestions. We hope that the revisions are satisfactory.

REVIEWERS' COMMENTS:

Reviewer #2 (Remarks to the Author):

The new Fig. 7 is helpful. However, it's unclear which exact analysis the significant "brain-to-brain" channel (CH3) came from, since there are so many analyses in the paper. Relatedly, this figure only shows homologous coupling (diagonal of the inter-subject correlation matrix), but there were significantly coupled off-diagonal channel pairs (such as Ch3-Ch7, shown in Figure 3A, I believe). Please clarify in the Figure 7 legend. Also, I would clean up the labels to be as clear as possible by changing to the following:

- "Significant brain-to-brain synchronization (speaker-listener)"
- "Significant brain-to-speech synchronization (listener's brain)"
- "Significant brain-to-speech synchronization (speaker's brain)"

Also:

Figure 1 is quite confusing. Why are the same blue and pink colors used to distinguish the listener and speaker and the source and detector, which are completely unrelated concepts? Simply change the speaker and listener color scheme. Also, I'd show Figure 1A from the perspective of the listener (on the bottom of the image), so that the speaker on the left is also on the reader's left (right now it's flipped and confusing). Also, the legend states that the angle between speakers was 60 degrees, but it is shown to be >90 .

Reviewer #3 (Remarks to the Author):

I am satisfied with the current version of the manuscript

Response to reviewers' comments

REVIEWERS' COMMENTS:

Reviewer #2 (Remarks to the Author):

The new Fig. 7 is helpful. However, it's unclear which exact analysis the significant "brain-to-brain" channel (CH3) came from, since there are so many analyses in the paper. Relatedly, this figure only shows homologous coupling (diagonal of the inter-subject correlation matrix), but there were significantly coupled off-diagonal channel pairs (such as Ch3-Ch7, shown in Figure 3A, I believe). Please clarify in the Figure 7 legend. Also, I would clean up the labels to be as clear as possible by changing to the following:

- "Significant brain-to-brain synchronization (speaker-listener)"
- "Significant brain-to-speech synchronization (listener's brain)"
- "Significant brain-to-speech synchronization (speaker's brain)"

Answer 1(A1): Figure 7 has been thoroughly changed to make these issues clear. Specifically, 1) a three-brain image as that in Figure 3b has been added to present significant brain-to-brain synchronization for both homologous (diagonal) and off-diagonal couplings. 2) CHs that showed significant brain-to-brain synchronization were colored in turquoise in order to distinguish them from CHs that showed significant brain-to-speech synchronization. 3) The analyses that generated these results are described in the figure legend.

In addition, CHs that showed significant brain-to-speech synchronization in the listener's brain were marked in yellow in the listener's brain, whereas those in the speakers' brain were marked in brown in the speakers' brain. The analyses that provided these results are also described in the figure legend.

Finally, the texts in Figure 7 were also modified according to Reviewer #2's above suggestions.

Also:

Figure 1 is quite confusing. Why are the same blue and pink colors used to distinguish the listener and speaker and the source and detector, which are completely unrelated concepts? Simply change the speaker and listener color scheme. Also, I'd show Figure 1A from the perspective of the listener (on the bottom of the image), so that the speaker on the left is also on the reader's left (right now it's flipped and confusing). Also, the legend states that the angle between speakers was 60 degrees, but it is shown to be >90 .

A2: The color scheme for the speaker and listener has been changed in Figure 1. Also, Figure 1a was modified so it is now from the perspective of the listener. Finally, the angle between the two speakers was modified to precisely show the 60 degrees.

Reviewer #3 (Remarks to the Author):

I am satisfied with the current version of the manuscript

A1: Thanks.

We thank both reviewers again for helping us improve the manuscript.